# LEARNING PRIMITIVE EMBODIED WORLD MODELS: TOWARDS SCALABLE ROBOTIC LEARNING

**Qiao Sun[1,2], Liujia Yang[1,3], Wei Tang[1,4], Wei Huang[1,5], Kaixin Xu[1,3], Yongchao Chen[6], Mingyu Liu[1,7], Jiange Yang[1,8], Haoyi Zhu[1,9], Yating Wang[1,10], Tong He[1], Yilun Chen[1], Xili Dai[11], Nanyang Ye[3], Qinying Gu[1]**

[1]Shanghai AI Lab   [2]Fudan   [3]SJTU   [4]NJUST   [5]THU
[6]Harvard   [7]ZJU   [8]NJU   [9]USTC   [10]Tongji   [11]HKUST (GZ)

https://qiaosun22.github.io/PrimitiveWorld/

## ABSTRACT

While video-generation-based embodied world models have gained increasing attention, their reliance on large-scale embodied interaction data remains a key bottleneck. The scarcity, difficulty of collection, and high dimensionality of embodied data fundamentally limit the alignment granularity between language and actions and exacerbate the challenge of long-horizon video generation–hindering generative models from achieving a *"GPT moment"* in the embodied domain. There is a naive observation: *the diversity of embodied data far exceeds the relatively small space of possible primitive motions*. Based on this insight, we propose **Primitive Embodied World Models** (PEWM), which restricts video generation to fixed shorter horizons, our approach *1) enables* fine-grained alignment between linguistic concepts and visual representations of robotic actions, *2) reduces* learning complexity, *3) improves* data efficiency in embodied data collection, and *4) decreases* inference latency. By equipping with a modular Vision-Language Model (VLM) planner and a Start-Goal heatmap Guidance mechanism (SGG), PEWM further enables flexible closed-loop control and supports compositional generalization of primitive-level policies over extended, complex tasks. Our framework leverages the spatiotemporal vision priors in video models and the semantic awareness of VLMs to bridge the gap between fine-grained physical interaction and high-level reasoning, paving the way toward scalable, interpretable, and general-purpose embodied intelligence.

## 1 INTRODUCTION

Embodied agents capable of planning and decision-making in complex environments rely heavily on internal representations of the world, commonly referred to as *world models*. Recent advances

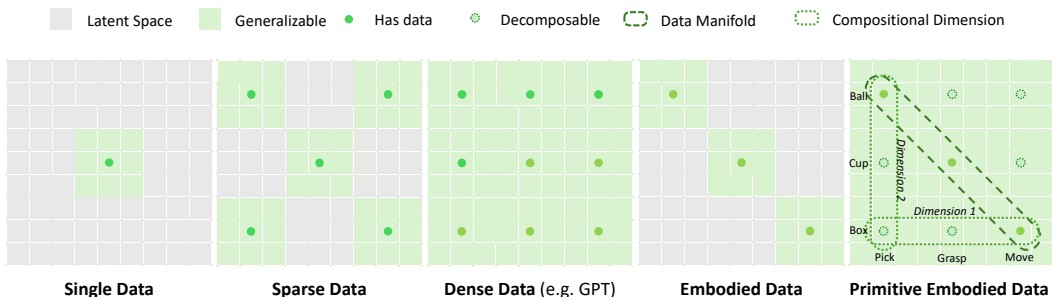

Figure 1: While densely distributed data can enable broad generalization, embodied data often suffer from sparsity (Du & Kaelbling, 2024; Xue et al., 2025b). The rightmost schematic highlights how organizing embodied data at the primitive level–along orthogonal dimensions such as action and object–supports compositional generalization even under limited data availability.

in pretrained video generation models–powered by self-supervised learning on vast internet-scale datasets–have demonstrated remarkable capabilities in synthesizing fine-grained, temporally coherent visual sequences from natural language descriptions (Du et al., 2023).

These successes have inspired a growing body of work to leverage such models as world models for embodied agents, casting policy learning as a video generation problem (Ko et al., 2023; Du et al., 2023; Yang et al., 2023; Brooks et al., 2024; Black et al., 2024a; Xiang et al., 2024; Bruce et al., 2024; Hu et al., 2024; Zhou et al., 2024; Dalal et al., 2025; Liang et al., 2024a; Clark et al., 2025; Hu et al., 2024; Li et al., 2025; Team et al., 2025; Pertsch et al., 2025; Zhen et al., 2025b;c;a; Zhu et al., 2025; Hafner et al., 2025): given a language-specified goal, the model generates a future video roll-out, from which actions are subsequently extracted. This paradigm offers compelling advantages in interpretability, generalization, and unified representation across diverse environments.

Despite this promise, two fundamental challenges remain largely unaddressed:

First, **we should rethink alignment before scaling**. Current approaches exhibit a trend toward ever-larger models and longer generation horizons, under *the assumption that extended prediction leads to better planning*. However, we argue that **such scaling is fundamentally constrained by the nature of embodied data**: the high dimensionality, sparsity, and collection difficulty of real-world interaction data severely limit the feasibility of long-horizon, high-fidelity video prediction. Moreover, fine-grained alignment between linguistic concepts and low-level actions becomes increasingly ill-posed as the prediction window grows.

Second, **data is the elepant in the room**. Training embodied world models critically depends on data–especially in a field where embodied intelligence itself lacks standardization, from sensor configurations to ontologies. Yet, most existing work focuses heavily on model optimization while largely overlooking the design of the underlying data. In practice, the gains from hastily training on a few fragmented, open-source datasets with highly scattered distributions are often limited. A more principled approach–co-designing the data strategy from the ground up in tandem with the model–can yield significantly better results. Figure 1 illustrates this insight through a heuristic analysis.

In this work, we challenge the prevailing long-horizon paradigm and propose a shift toward a promising **primitive-level modeling**. We introduce **Primitive Embodied World Models (PEWM)**, a new paradigm that restricts video generation to short, fixed-length horizons.

By focusing on predicting immediate, primitive-scale transitions, PEWM enables (1) fine-grained alignment between language and action, (2) reduced modeling complexity, (3) improved data efficiency in collection and training, and (4) lower inference latency–effectively unlocking the world models potential as a *"cerebellum"* for fast, reactive control.

Our framework bridges the gap between low-level *"cerebellum"*-like dynamics modeling and high-level *"cerebral cortex"*-inspired planning, paving the way toward scalable, interpretable, and truly general embodied intelligence. We demonstrate PEWMs effectiveness in simulation and real-robot experiments, showing strong generalization to novel instructions, robustness to domain shifts, and efficient adaptation with minimal task-specific data.

## 2 PRIMITIVE EMBODIED DATA PAVING THE WAY TOWARDS SCALABLE EMBODIED LEARNING

### 2.1 THE MOTIVATION AND DEFINITION OF PRIMITIVE EMBODIED DATA

> *"What can be said at all can be said clearly, and what we cannot talk about we must pass over in silence."*
>
> —— Ludwig Josef Johann Wittgenstein

Collecting real-world embodied data is labor-intensive, and even large-scale datasets suffer from sparsity, high dimensionality, and limited generalization beyond minor scene variations (Xue et al., 2025b; Brohan et al., 2022; Collaboration et al., 2023; Khazatsky et al., 2024; Walke et al., 2023; Dalal et al., 2025), making fine-grained language-action alignment especially challenging. To overcome this, we propose organizing embodied experience at the level of *primitives*–short, language-

**Human Input:** Pick the yellow tape measure ⇒ P1 → P2 → P3

**P1: Move** the **gripper** to the **yellow tape measure**

**P2: Close** the **gripper** *(directly execute)*

**P3: Lift** the **gripper**

Figure 2: Illustration of primitive-level task execution for "Pick up the yellow tape measure." 6-DoF motions are meant to be rolled out via diffusion, while discrete gripper actions are handled directly through symbolic execution. Note that this is a simple task, chosen for ease of illustration.

grounded action units that serve as atomic building blocks for complex behaviors, enabling denser supervision and scalable compositional generalization.

**Definition 2.1** (Primitive as Semantically Atomic Action Unit)**.** A *primitive* is a finite-duration embodied trajectory $p = (\mathbf{x}_{0:T}, a_{0:T-1})$, paired with a natural language instruction $u$, such that:

1. **Semantic atomicity**: $u$ expresses a single, coherent manipulation intent that cannot be meaningfully decomposed into shorter language-grounded sub-instructions without loss of task-level meaning;
2. **Temporal locality**: $T$ is short enough (e.g., $\leq 2$ seconds) to support high-fidelity video generation and precise 6-DoF trajectory extraction;
3. **Generative feasibility**: Given an initial observation and spatial guidance (e.g., startgoal heatmaps), a diffusion-based world model can reliably generate a plausible visual rollout of $p$.

Crucially, a primitive is **NOT** defined by mechanical simplicity, but by semantic indivisibility under language grounding. Thus, while pick cup qualifies, so does arrange flowers–provided it is executed as a unified intent within a short horizon and describable by a single instruction.

Building on this notion, we posit that complex tasks are inherently compositional:

**Assumption 2.2** (Compositional Decomposability of Embodied Tasks)**.** Any long-horizon embodied task can be decomposed into a finite sequence of one or more primitives, each satisfying Definition 2.1.

This leads to a key structural insight:

**Corollary 2.3** (Compact Primitive Template Basis)**.** *The number of distinct primitive templates (e.g., "pick", "open", "arrange") is vastly smaller than the total number of possible embodied trajectories, due to the combinatorial explosion of object, scene, and embodiment configurations.*

This disparity implies that primitive-centric data organization is not merely convenient–it is fundamentally more efficient and scalable. Specifically, it enables:

1. **High data efficiency**: A single long-horizon demonstration yields multiple labeled primitives, dramatically increasing the effective sample count and mitigating data sparsity (Fig. 1);
2. **Structured low-dimensional learning**: Each primitive is short, visually coherent, and aligned with a single linguistic concept, facilitating fine-grained alignment between language and action in the video world model;
3. **Direct action extraction**: High-fidelity, short-horizon video generation enables zero-shot 6-DoF trajectory extraction (Appendix Sec. B), eliminating the need for task-specific policy heads;
4. **Plug-and-play compositional generalization**: A trained primitive world model becomes a reusable "skill library" that a high-level VLM planner can sequence on-the-fly (Fig. 2), supporting zero-shot execution of novel long-horizon tasks without retraining.

In sum, primitive embodied data provides a principled interface between symbolic reasoning (language) and continuous dynamics (video), turning the world model into a modular, interpretable, and scalable cerebellum for embodied intelligence.

## 2.2 DATA COLLECTION

**Enhancement in Efficiency, Density, and Quality** We construct a primitive-centric dataset $\mathcal{D}_{\text{prim}} = \{E_i\}$, where each episode $E_i$ is decomposed into primitives $\{\mathcal{P}^k\}_{k=1}^{N_i}$ with $\mathcal{P}^k = (\mathbf{x}_{t^k}^{\text{img}}, \text{Instr}^k, H_{s \to g}^k, \mathbf{x}_{t^k+1:t^{k+1}-1}^{\text{img}})$. Here, $\mathbf{x}_{t^k}^{\text{img}}$ is the initial image, $\text{Instr}^k$ the instruction, $H_{s \to g}^k$ the start-goal heatmap, and future frames $\mathbf{x}_{t^k+1:t^{k+1}-1}^{\text{img}}$ are to be predicted.

To enhance data collection efficiency, density, and quality, we introduce improvements along both *temporal* and *spatial* dimensions: 1) *Spatially*, five synchronized cameras capture each $\mathcal{P}^k$ in parallel, increasing **data density** and **spatial coverage**. All 10K+ episodes are collected by co-authors, ensuring **high quality** and **consistency**. 2) *Temporally*, we encode primitive boundaries $\{t^k\}$ via teleoperation device buttons, enabling on-the-fly segmentation. This yields 5.8 primitives per session on average, together with spatial efficiency, boosting **collection efficiency** by up to 29×.

We use Qwen2.5-VL 7B for few-shot pre-annotation of $\text{Instr}^k$, correct a 10% subset, then fine-tune the model to annotate the remainder (see Appendix D.1). Unlike methods with fixed primitive grammars, our approach is soft and on-the-fly, enabling flexibility, openness, and scalable annotation.

**Full-Arm Shotting Facilitating Learning-Based Embodied World Simulator** Another notable design choice in $\mathcal{D}_{\text{prim}}$ is that the full robot arm–including the base and joint structures–is meant to be visible in the field of view, in contrast to cropped, end-effector-only frames common in prior datasets. This enables the diffusion model to internalize soft physical constraints such as reachability, joint limits, and spatial feasibility during prediction.

This holistic visual representation, combined with multi-view observations, allows the world model to function as a **learning-based generative simulator**. By observing the same robotic action from multiple calibrated viewpoints, the model learns a more robust and view-invariant latent dynamics space, where transitions are consistent across perspectives. This enables the model to implicitly align action effects across views with natural language descriptions, capturing 3D spatial relationships and occlusion patterns without explicit geometric supervision. As a result, the model not only simulates plausible future states but also generalizes better to novel viewpoints and scene configurations–key properties of a reliable, data-driven simulator.

**The High Potential of Leveraging Action-Free Video Data for Scalable, Web-Scale Embodied Learning** It is worth noting that the LIBERO (Liu et al., 2024a) data we use not only includes the original dataset's multi-view replays but also videos generated through OpenVLA rollouts as training data. This directly demonstrates our methods ability to leverage any embodied video–whether from rollouts of other policies or from web-scale data–to improve model performance, highlighting one of the key advantages of using video as a universal representation.

## 3 LEARNING PRIMITIVE EMBODIED WORLD MODELS

### 3.1 DUAL-LEVEL COMPOSITIONAL GENERALIZATION

Our method enhances generalization along two orthogonal axes: (1) *intra-primitive compositional generalization*, where diffusion models trained on densely annotated, semantically rich primitives learn to recombine fine-grained visual and dynamic elements (e.g., shape, motion, spatial relations) in novel ways; and (2) *inter-primitive combinatorial generalization*, achieved by flexibly sequencing primitives to generate complex, realistic behaviors that far exceed the complexity of individual primitives.

**Implicit Compositionality of Diffusion Video Generation** Diffusion models are capable of generating photo-realistic images that combine elements which likely do not appear together in the training set, demonstrating the ability to compositionally generalize (Liu et al., 2022a; Liang et al., 2024b).

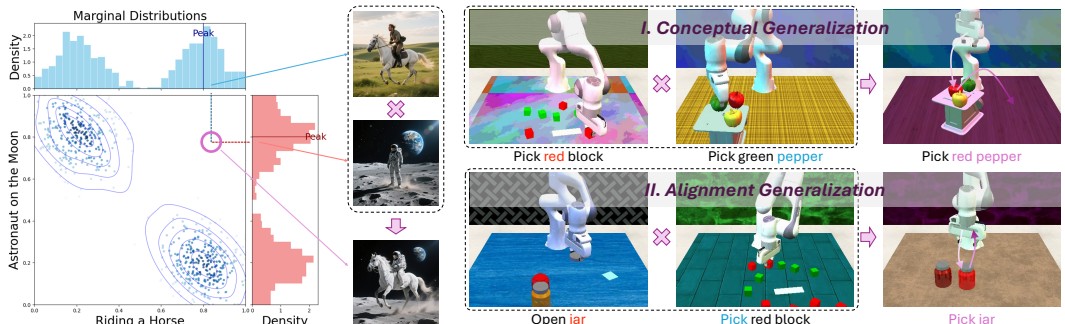

Figure 3: An analogy to highlight the compositional generalization capability of our approach.

We draw an analogy to highlight the compositional generalization capability of our approach. On the left side of Figure 3, we show a classic example from 2D vision: the well-known "astronaut riding a horse" image, where a model combines two rarely co-occurring concepts into a coherent and recognizable scene–demonstrating strong compositional understanding in static images. On the right, we present a corresponding case in embodied video generation: although our model has only observed "pick apple" and "open jar" separately during training, it can naturally generalize to the unseen combination "pick jar".

We ground this compositional generalization in an Energy-Based Model (EBM) perspective. EBMs naturally support compositionality through additive energy decomposition: $E(\mathbf{x}) = \sum_i E_i(\mathbf{x}; \phi_i)$, where each term $E_i$ corresponds to a semantic factor (e.g., object, action, or spatial relation). Novel combinations arise by recombining known factors via energy minimization.

Although our diffusion model learns the score function $\nabla_{\mathbf{x}} \log p(\mathbf{x})$, it implicitly defines an energy landscape $E(\mathbf{x}) = -\log p(\mathbf{x})$ with the same modular structure. During generation, activations from previously disjoint patterns (e.g., pick and jar) combine to produce coherent, unseen behaviors–enabling zero-shot generalization through factorized semantic priors.

Given the diverse and realistic co-occurrence of embodied concepts–such as shape, motion, and spatial relations–in our training data, the model learns rich semantic priors within each 32-frame primitive, enabling strong compositional generalization. Spatiotemporal coherence is inherited from large-scale video pretraining, a key advantage over static image-based methods. We treat each primitive as a distribution and use a heatmap as conditional input, formulating generation as: $\mathbf{x}_{1:T}^{\text{img}} \sim P(\mathbf{x}_{1:T}^{\text{img}} \mid H_{s \to g}, \text{img}_0)$. We adopt an I2V model as the world model $\mathcal{W} : (\mathbf{x}_0^{\text{img}}, H_{s \to g}) \mapsto \mathbf{x}_{1:T}^{\text{img}}$. The base model is DynamiCrafter (Xing et al., 2024), which is pre-trained on data without robotic arms. Our method performs well on robotic tasks, demonstrating effective zero-shot transfer. See Appendix F.1 for details.

**Explicit Compositionality by Sequentially Combining Primitives**    By sequentially composing primitives, we treat the Primitive-Enabled World Model (PEWM) as a plug-and-play module for constructing complex behaviors. Each primitive $\pi_i \in \mathcal{P}$ represents a reusable, low-level dynamic policy that maps a current image $\mathbf{x}_0$ and a goal heatmap $h_i$ to a future video segment: $\mathbf{x}_{1:T_i} \sim \mathcal{W}(\mathbf{x}_0, h_i; \pi_i)$, where $\mathcal{W}$ is the world model. Chaining $N$ primitives yields a full trajectory: $\mathbf{x}_{1:T} = \text{Compose}(\{\mathcal{W}(\mathbf{x}_{t_{i-1}}, h_i; \pi_i)\}_{i=1}^N)$, enabling long-horizon, high-level behaviors through explicit, interpretable composition. This design supports systematic generalization–novel sequences can be formed from known primitives, even if never seen during training.

Notably, the mapping from instruction $u$ to primitive sequence $\{\pi_i\}_{i=1}^N$ is learned from data collected during our annotation pipeline. The model implementing this mapping–denoted as $\mathcal{P}_{\text{LoRA}}(u) \to \{\pi_i\}$–is shared with the LoRA-based planner in the VLM planner module, significantly reducing training overhead. This same model is used to auto-label trajectories throughout our workflow: we iteratively generate pseudo-labels with $\mathcal{P}_{\text{LoRA}}$, refine them manually, and retrain $\mathcal{P}_{\text{LoRA}}$ on the improved dataset–enabling continuous self-improvement.

## 3.2 TRAINING STRATEGY

**Sim-Real Hybrid Data Strategy** In practice, we found that when training solely on real-world data, the model struggles to capture fine details of fast-moving parts such as the gripper, due to the reconstruction loss in the VAE. To enable the model to learn cleaner proprioceptive motion patterns, we introduced simulation data, including data from RLBench (James et al., 2020a) and LIBERO (Liu et al., 2024a), and adopt a sim-real hybrid data mixing strategy. For detailed data information, please refer to Appendix D.1. This approach proves highly effective: the final model successfully combines the rich, complex textures from real-world data with the precise and clear kinematic motions from simulation, significantly enhancing the overall generation quality. Appendix I.3 Figure 12 shows a comprehensive comparison.

**Three-Stage Finetuning Base Video Generation Model on Primitive Embodied Data** We build upon the strong pretraining of large-scale video generation models. In this work, we introduce a three-stage fine-tuning strategy to adapt the model to primitive-based embodied tasks, balancing simulation-to-reality transfer and semantic alignment: *Stage 1: Simulation Pre-Finetuning.* We first fine-tune the model on simulated data with a low number of epochs and early stopping, halting once generated videos exhibit plausible motion trends (e.g., arm movement toward an object). This rapid adaptation injects embodied semantics–such as robot dynamics, spatial interactions, and action semantics–into the model while avoiding premature convergence to simulation-specific modes. *Stage 2: Balanced Simulation-Real Mixing.* We reduce the learning rate by half and train on a 1:1 mixture of real and simulated data. This stage encourages the model to align dynamics and appearance across domains, producing videos with clear, consistent actions and object structures in both settings. *Stage 3: Reality-Centric Refinement.* Finally, we shift the data ratio to 80% real and 20% simulated, focusing the model on high-fidelity real-world generation while retaining the broad coverage of simulation. This stage emphasizes precise primitive-level temporal structure and fine-grained visual details. For further training details (e.g., hyperparameters, augmentation, and evaluation criteria), please refer to Appendix F.2.

## 3.3 CAUSAL DISTILLATION AND ACCELERATION: REAL-TIME ON-THE-FLY FUTURE FRAME PREDICTION

Real-time performance is critical in robotic learning–systems operating at low or irregular frequencies struggle to react to dynamic environments. This poses a significant challenge for diffusion-based video generation, where traditional models generate entire videos (or full latent sequences) non-causally, denoising over many steps and producing all frames only after a long latency. Such autoregressive-in-time but non-causal-in-context generation is incompatible with real-time deployment.

To address this, we adopt *causal video generation* via knowledge distillation, following Self Forcing (Yin et al., 2025). We train a student model to predict future frames in a strictly causal manner–each frame is generated based on past observations only, without access to future context. The student performs only **4 denoising steps** per chunk (with 4 frames), enabling low-latency inference. To maintain generation quality under aggressive acceleration, we employ *self-forcing*–using the models own past predictions as input for future steps, closing the loop between prediction and conditioning.

As a result, our system achieves real-time on-the-fly prediction at **12 FPS** on standard hardware, making it suitable for closed-loop robotic control. This enables the world model to provide timely visual predictions that align with actual robot execution frequency.

# 4 APPLICATIONS

## 4.1 APPLICATION 1: HIGH-QUALITY EMBODIED VIDEO GENERATION FACILITATING DIRECT 6-DOF TRAJECTORY EXTRACTION

By integrating the above techniques–curated data design, staged fine-tuning, causal distillation, and closed-loop rollout–our approach achieves strong video generation performance using a relatively small model (1.4B parameters).

Prior methods (Du et al., 2023; Zhen et al., 2025a) often require a learning-based policy head or inverse kinematics (IK) module–after video generation, due to insufficient spatiotemporal precision. In contrast, we argue that our method is the first to make it practically feasible to **directly extract end-effector 6-DoF trajectories from generated videos without any task-specific adaptation**, as illustrated in Figure 4. This not only demonstrates the high spatiotemporal precision of our method, but also eliminates the need for additional learning or task-specific configurations.

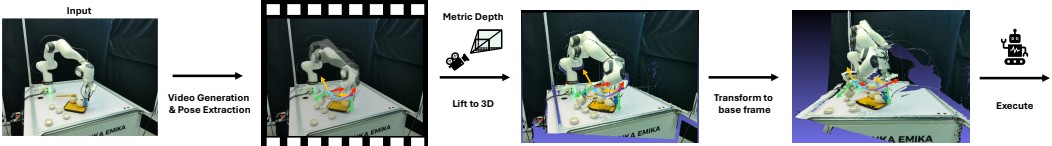

Figure 4: Direct 6-DoF end-effector trajectory extraction from generated videos.

We apply Gen6D (Liu et al., 2022b), an off-the-shelf, zero-shot 6-DoF pose estimator, to the generated video frames to recover the full 6-DoF motion trajectory of the robot end-effector. This enables direct mapping from visual prediction to actionable control signals–bridging the gap between generative world models and real robot execution. Technical details of this extraction pipeline are provided in Appendix B. The experimental results are provided in Appendix B.2.

## 4.2   APPLICATION 2: PLUG-AND-PLAY LONG-HORIZON COMPOSITIONAL GENERALIZATION

In the previous section, we described open-loop future prediction via single-step rollout. In this section, we close the loop by feeding the models generated future frames back as input for subsequent predictions, enabling iterative planning and execution: $\mathbf{x}_{t+1:T}^{(k)} \sim \mathcal{W}(\mathbf{x}_t, h^{(k)}), \quad \mathbf{x}_t \leftarrow \text{crop}(\mathbf{x}_t^{(k)})$, where $k$ denotes the current planning step, and the cropped latest frame from the generated rollout becomes the new input $\mathbf{x}_t$ for the next prediction. This autoregressive, closed-loop deployment allows the world model to continuously adapt to the actual environment state, correcting for drift and disturbances. Figure 5 presents the pipeline of this hierachical long-horizon compositional generalization. Appendix C presents a more detailed workflow.

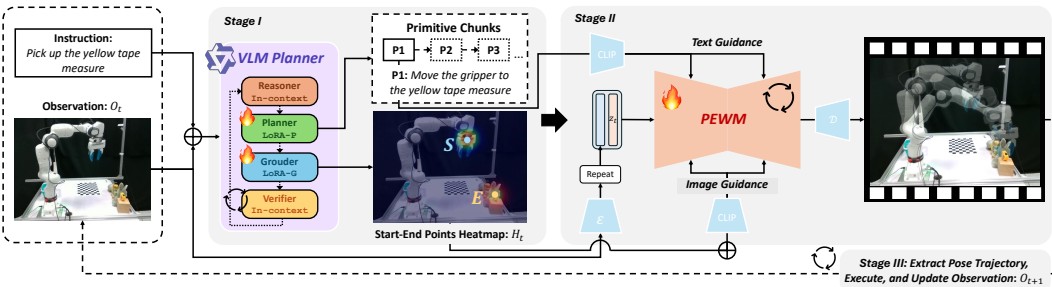

Figure 5: Closed-loop, autoregressive planning via iterative rollouts. The model feeds generated frames back as input, enabling continuous adaptation and long-horizon control.

Table 1 presents a comprehensive comparison of overall success rates across nine manipulation tasks from the RLBench benchmark . The performance of our method is assessed against several existing baselines, including Image-BC, UniPi, and 4DWM, with success rates averaged over 100 episodes.

## 4.3   APPLICATION 3: AS DATA SYNTHESIS ENGINES

Beyond policy execution, our Primitive World Model serves as a scalable data synthesis engine. By conditioning on language instructions and initial visual states, it generates physically plausible, high-fidelity video rollouts with consistent robot dynamics–enabling data augmentation for rare interactions, failure modes, or scene variations (e.g., lighting, background). Trained on hybrid sim-real data and full-arm observations, the model internalizes soft embodiment constraints, yielding videos with strong physical consistency (EPiCS: 11.45/13, Table 12). This makes the generated data more

Table 1: **Overall success rate on RLBench tasks.** We compare our method against existing baselines on 9 manipulation tasks from the RLBench benchmark (James et al., 2020b), with success rates averaged over 100 episodes. Results for Image-BC (Jang et al., 2022), UniPi (Du et al., 2023), and 4DWM (Zhen et al., 2025a) are directly cited from the 4DWM paper. Our method (bottom row) achieves the highest success rate on most tasks, with consistent gains in articulated and contact-rich scenarios.

| Methods | close box | open drawer | open jar | open microwave | put knife | sweep to dustpan | lid off | weighing off | water plants |
|---------|-----------|-------------|----------|----------------|-----------|------------------|---------|--------------|--------------|
| Image-BC | 53 | 4 | 0 | 5 | 0 | 0 | 12 | 21 | 0 |
| UniPi | 81 | 67 | 38 | _72_ | 66 | 49 | _70_ | **68** | 35 |
| 4DWM | _88_ | _80_ | **44** | 70 | _70_ | _56_ | **73** | _62_ | _41_ |
| **Ours** | **93** | **84** | _43_ | **78** | **72** | **63** | 67 | 58 | **56** |

effective for sim-to-real transfer than graphics-based simulation, turning the world model into a force multiplier for imitation and reinforcement learning.

For *more promising applications*, please see Section 7 and Appendix L.

## 5 ANALYSIS

**Performance    1) Planning.** Our method achieves high planning accuracy on unseen tasks by decomposing them into spatially grounded primitives. As shown in Table 2, it attains 18/20 primitive accuracy on pick up cup, 16/20 on move cloth, and 15/20 on fold cloth. In contrast, OpenVLA fails completely in zero-shot (0/20), underscoring its dependence on task-specific fine-tuning. Our approach generalizes without retraining the video model, thanks to its modular design and explicit spatial grounding. **2) Primitive Execution.** The integration of planning and video generation translates into strong execution: our method achieves 16/20 task success on cup picking, versus 12/20 for OpenVLA; similarly, 14/20 vs. 10/20 for moving cloth, and 13/20 vs. 4/20 for folding cloth. Success on these out-of-distribution tasks demonstrates robust generalization, enabled by precise primitive segmentation and high-fidelity video rollouts. **3) Long-Horizon Tasks.** We handle long-horizon tasks by decomposing them into short-horizon, plug-and-play primitiveseach aligned with the contextual scope of vision-language models (VLMs). This mitigates drift and inconsistency in extended planning. As shown in Appendix I.4 (Fig. 13), generated video frames are projected to real-world coordinates via camera intrinsics/extrinsics, enabling accurate pose estimation per primitive while preserving global task coherence.

Table 2: **Performance breakdown on three real-world tasks across planning, video generation, and primitive execution.** "Ours" denotes our method with frozen video diffusion; "OpenVLA" represents a strong end-to-end baseline. We additionally report OpenVLA's zero-shot performance without task-specific fine-tuning.

| Task | Stage | Metric | Ours | OpenVLA | OpenVLA (ZS) |
|------|-------|--------|------|---------|--------------|
| Pick up cup | Planning | Primitive accuracy | 18 / 20 | N/A | N/A |
| | Video Generation | Frame realism (✓/total) | 17 / 20 | N/A | N/A |
| | Primitive Execution | Task success | **16** / 20 | 12 / 20 | 0 / 20 |
| Move cloth | Planning | Primitive accuracy | 16 / 20 | N/A | N/A |
| | Video Generation | Frame realism (✓/total) | 15 / 20 | N/A | N/A |
| | Primitive Execution | Task success | **14** / 20 | 10 / 20 | 0 / 20 |
| Fold cloth | Planning | Primitive accuracy | 15 / 20 | N/A | N/A |
| | Video Generation | Frame realism (✓/total) | 14 / 20 | N/A | N/A |
| | Primitive Execution | Task success | **13** / 20 | 4 / 20 | 0 / 20 |

**Primitive-level Compositional Generalization**    A core strength of our primitive-centric framework is its ability to achieve *conceptual compositional generalization* (Figure 3)–the capacity to succeed on novel tasks that were never explicitly demonstrated during training. We quantitatively evaluate this capability in Table 3, which reports success rates for executing unseen (predicate, object) combinations on a real robot. The results demonstrate strong generalization: for instance, the "pick" primitive, trained on objects like cups and boxes, successfully generalizes to the novel combination "pick jar" with an 80% success rate. Similarly, the "push" action generalizes effectively to cups and jars, despite these pairings being absent from the training data. Even for the more com-

plex "open" action, the model transfers successfully to the unseen "open cup" task (70% success), showcasing its ability to adapt a skill to an object with significantly different geometry and affordances. This generalization is enabled by two key factors. (1) Our video diffusion model, trained on densely annotated primitive data, learns rich, disentangled representations of action dynamics and object properties. This allows it to condition the generation of a "pick" motion on the visual features of a previously unseen jar. (2) The VLM-based primitive planner provides semantic and spatial grounding, correctly identifying the target object and generating a feasible subgoal configuration even for unseen objects. The high success rates on these zero-shot combinations underscore that our approach moves beyond simple imitation learning, instead capturing the underlying compositional structure of embodied manipulation, a crucial step toward truly flexible and scalable robotic intelligence.

Table 3: Primitive-level compositional generalization: success rates (successes / 10) for unseen (predicate, object) pairs. Rows = predicates, columns = objects. Bold cells indicate compositions that were **not** present during training.

| Predicate \ Object | cup | box | drawer | jar |
|---|---|---|---|---|
| pick | 9/10 | **8/10** | – | **8/10** |
| open | – | 9/10 | 9/10 | 7/10 |
| push | **7/10** | **8/10** | 8/10 | **6/10** |

**Efficiency**     Our method is significantly more efficient than large diffusion baselines (e.g., Hunyuan I2V, Wan 2.1), achieving up to 75x faster inference and 67x lower VRAM usage (Appendix Table 9). This enables real-time deployment without sacrificing video quality or task relevance.

**Ablation Study**     We ablate two core components: (1) the VLM-based primitive planner, and (2) simulation-augmented video training. Results in Table 4 show that removing Start-Goal Guidance reduces success rates (e.g., from 16/20 to 12/20 on cup picking), confirming the value of explicit spatial grounding. (3) Eliminating primitive decomposition entirelymapping instructions directly to actionscauses severe drops (e.g., 3/20 on cloth folding), highlighting the necessity of structured planning. Similarly, training the video model on real data alone (without simulation) degrades performance across all tasks, verifying that simulated data enhances generalization to real-world object configurations.

Table 4: **Ablation study on model components.** Task success rate is reported over 20 trials per setting. "Primitive planner ablations" isolate the effect of VLM-based spatial grounding; "VideoGen ablations" evaluate the impact of simulation-augmented training.

| Ablation Group | Pick up cup | Move cloth | Fold cloth |
|---|---|---|---|
| Full model | **16 / 20** | **14 / 20** | **13 / 20** |
| w/o SGG | 12 / 20 | 10 / 20 | 7 / 20 |
| w/o Primitive Planner | 9 / 20 | 5 / 20 | 3 / 20 |
| w/o Sim data | 12 / 20 | 9 / 20 | 5 / 20 |

## 6   RELATED WORK

**Video Generation as World Models.** Diffusion models have become dominant in image generation (Ho et al., 2020; Song et al., 2021; Rombach et al., 2022b), and their extension to videovia 3D convolutions, latent spaces, or spatiotemporal attentionhas yielded strong generative performance (He et al., 2023; Blattmann et al., 2023; Singer et al., 2022; Chen et al., 2024a). The success of Sora (Brooks et al., 2024; Liu et al., 2024e) has further established Transformer-based video diffusion (e.g., DiT (Peebles & Xie, 2023)) as a leading paradigm, enabling improved temporal coherence and scalability (Zheng et al., 2024; Yang et al., 2024; Kong et al., 2024). These models are increasingly viewed as pixel-level world modelssystems that predict future observations given actions and context (Ha & Schmidhuber, 2018; Hafner et al., 2020; Micheli et al., 2022). Recent works frame video generation itself as world modeling (Ko et al., 2023; Du et al., 2023; Bruce et al., 2024; Zhou et al., 2024; Li et al., 2025), often using large DiT-based architectures trained under unified generative objectives. However, such models are computationally expensive and lack real-time

responsiveness. Hierarchical alternatives (Ajay et al., 2023; Dalal et al., 2025; Zhou et al., 2024) improve structure but suffer from error propagation and rigid cross-modal alignment. Crucially, none integrate vision-language reasoning, planning, and grounding in a zero-shot policy execution frameworkkey distinctions of our approach.

**End-to-End Vision-Language-Action Learning.** Vision-Language-Action (VLA) models bypass explicit dynamics modeling by directly mapping multimodal inputs to actions (Brohan et al., 2023b; Kim et al., 2024a; Bharadhwaj et al., 2023). Leveraging pre-trained vision-language models (VLMs) (Chen et al., 2024c; Li et al., 2024; Beyer et al., 2024), they unify perception, grounding, and control in a single architecture. Action prediction is typically implemented via regression (Brohan et al., 2023a; Liu et al., 2024c), diffusion (Black et al., 2024b; Liu et al., 2024d), or hybrid schemes (Liu et al., 2025a; Ye et al., 2024). While effective on short-horizon tasks, these systems struggle with long-horizon generalization and cross-domain transfer. Recent efforts introduce hierarchyvia modular perception-policy stacks (Liu et al., 2024b; Bjorck et al., 2025) or skill decomposition (Shi et al., 2025; Raj et al., 2024)to improve scalability and interpretability. Yet, nearly all rely on in-domain demonstrations, limiting zero-shot adaptability. Our method circumvents this dependency by generating task-agnostic, primitive-level trajectories in simulation, enabling zero-shot policy execution without task-specific fine-tuning.

**Datasets for Robotic Learning.** Large-scale robotic datasetscollected via teleoperation (Brohan et al., 2022; Khazatsky et al., 2024), scripted policies (Collaboration et al., 2023; Gu et al., 2023), or expert rollouts (Schiavi et al., 2023)have been consolidated in benchmarks like OpenX-Embodiment (Collaboration et al., 2023). Despite scale, they remain costly to expand and limited in semantic or morphological diversity. RH20T-p (Chen et al., 2024b) adds primitive annotations but introduces noise; simulated environments (e.g., RLBench (James et al., 2020b), CALVIN (Mees et al., 2022)) offer control but lack realism. Recent works augment data via simulation or generative rendering (Mandlekar et al., 2023; Yang et al., 2025; Xue et al., 2025a), yet still depend on human demonstrations. In contrast, our framework synthesizes diverse, primitive-level trajectories de novo in simulationwithout human supervision or segmentationenabling efficient sim-to-real transfer when paired with pre-trained VLMs.

# 7 CONCLUSION, LIMITATIONS, AND FURTHER VISION

We propose a novel perspective on embodied world modeling: using unlabeled action videos as a universal representation, we segment long-horizon trajectories into semantically indivisible primitives–short clips aligned with atomic language instructions. This shift unlocks three key advantages: (1) Fine-grained languagevision alignment enables high-fidelity, efficient video generation with lower compute; (2) Primitive-centric data is lower-dimensional, denser, and easier to collect, while diffusion models provide strong intra-primitive compositional generalization; (3) A VLM planner can sequence these primitives via Start-Goal Heatmap Guidance (SGG), achieving inter-primitive building-block generalization–turning the world model into both a data engine and a plug-and-play policy backbone.

Our method outperforms imitation learning, vanilla video diffusion, VLAs, and state-of-the-art embodied world models by combining modular, primitive-level modeling with self-forcing distillation for real-time, low-step generation (12 FPS in 4 steps), zero-shot compositional generalization, and sim-to-real scalability–all while remaining interpretable, adaptive, and free from costly demonstrations or black-box policies, making it ideal for real-world robotic deployment.

While our framework marks a leap in real-time, interpretable robotic control, key frontiers remain: evolving toward a unified perception-action system that plans in latent space without language, pushing latency below 4-step inference for high-frequency tasks, and scaling to multi-agent, deformable-object, and dual-arm scenarios. Beyond model improvements, we call for standardized benchmarks to advance the field–positioning our primitive-centric, sim-real hybrid approach as a foundational paradigm for scalable, general-purpose embodied intelligence.

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

# APPENDIX

## A  THEORETICAL JUSTIFICATION FOR PRIMITIVE-CENTRIC EMBODIED LEARNING

We provide a mathematically grounded justification for the feasibility and expressivity of modeling embodied intelligence via a finite set of semantic primitives. Our derivation avoids pathological assumptions about trajectory indecomposability and instead builds upon three empirically plausible premises.

### A.1  FOUNDATIONAL ASSUMPTIONS

Let $X \subset \mathbb{R}^d$ be a compact metric space representing the robots observable state (e.g., RGB-D image, proprioception). Let $\mathcal{U}$ be a countable set of natural language instructions describing manipulation intents (e.g., "pick cup", "arrange flowers").

We assume:

**Assumption A.1** (Semantic Grounding). There exists a surjective mapping $\Phi : \mathcal{U} \to \mathcal{T}$, where $\mathcal{T} = \{\tau_1, \ldots, \tau_K\}$ is a **finite** set of *semantic primitive templates*, such that each $\tau_k$ corresponds to a reusable skill (e.g., "grasp", "place", "rotate") that generalizes across objects and scenes.

**Assumption A.2** (Temporal Locality). Each execution of a primitive template $\tau_k$ under instruction $u = \Phi^{-1}(\tau_k)$ yields a trajectory $p \in C([0, T_k], X)$ with bounded duration $T_k \leq T_{\max}$ (e.g., 2 seconds), and the mapping from initial state $x_0$ and goal $g$ to $p$ is continuous.

**Assumption A.3** (Compositional Execution). Any long-horizon task $U \in \mathcal{U}^*$ (finite sequence of instructions) induces a trajectory $\xi \in C([0, T], X)$ that can be written as a concatenation:

$$\xi = p_1 \cdot p_2 \cdot \cdots \cdot p_N,$$

where each $p_i$ is an instance of some $\tau_{k_i} \in \mathcal{T}$, grounded via perception.

### A.2  CONSTRUCTIVE APPROXIMATION THEOREM

Let $\mathcal{M}_T(X) = \{\xi \in C([0, T], X) \mid T > 0\}$ be the space of all finite-horizon continuous trajectories, equipped with the uniform metric:

$$d_\infty(\xi_1, \xi_2) = \sup_{t \in [0, T]} \|\xi_1(t) - \xi_2(t)\|.$$

Let $\mathcal{C}(\mathcal{T}) \subset \mathcal{M}(X)$ denote the set of all trajectories obtainable by finite composition of primitive instances from $\mathcal{T}$.

**Theorem A.4** (Density of Primitive Compositions). *Under Assumptions A.1–A.3, for any $\xi^* \in \mathcal{M}_T(X)$ and any $\epsilon > 0$, there exists a composed trajectory $\xi \in \mathcal{C}(\mathcal{T})$ such that $d_\infty(\xi, \xi^*) < \epsilon$.*

*Proof.* Since $X$ is compact and $\xi^*$ is continuous on $[0, T]$, it is uniformly continuous. Thus, for any $\epsilon > 0$, there exists $\delta > 0$ such that $|t - s| < \delta \Rightarrow \|\xi^*(t) - \xi^*(s)\| < \epsilon/2$.

Partition $[0, T]$ into $N$ intervals $[t_i, t_{i+1}]$ with $t_{i+1} - t_i < \min(\delta, T_{\max})$. For each segment $\xi_i^* = \xi^*|_{[t_i, t_{i+1}]}$, define a local intent $u_i$ that describes the transition from $\xi^*(t_i)$ to $\xi^*(t_{i+1})$ (e.g., "move end-effector from A to B").

By Assumption A.1, $u_i$ maps to some $\tau_{k_i} \in \mathcal{T}$. By Assumption A.2, there exists a primitive instance $p_i$ (grounded via SGG or VLM) such that $p_i(0) = \xi^*(t_i)$ and $\|p_i(t_{i+1} - t_i) - \xi^*(t_{i+1})\| < \epsilon/2$. By continuity of $p_i$ and $\xi_i^*$, and since both are defined on an interval of length $< \delta$, we have:

$$\sup_{t \in [t_i, t_{i+1}]} \|p_i(t - t_i) - \xi^*(t)\| < \epsilon.$$

Concatenating $\{p_i\}_{i=1}^N$ yields $\xi \in \mathcal{C}(\mathcal{T})$ with $d_\infty(\xi, \xi^*) < \epsilon$. $\square$

### A.3 Cardinality Disparity and Data Efficiency

Let $|\mathcal{T}| = K \ll \infty$. The number of possible composed trajectories of length $N$ is at most $K^N$, which is **countable**. However, $\mathcal{M}_T(X)$ has cardinality $2^{\aleph_0}$ (uncountable). Despite this, Theorem A.4 shows that a **countable** set $\mathcal{C}(\mathcal{T})$ is **dense** in an **uncountable** spaceexactly analogous to $\mathbb{Q}$ being dense in $\mathbb{R}$.

This implies:

1. **Data efficiency**: Learning $K$ primitives yields coverage of a dense subset of behaviors.
2. **Generalization**: Novel tasks are approximated by novel compositions, not novel primitives.
3. **Scalability**: Adding new objects/scenes requires no new primitive templatesonly new grounding.

Thus, primitive-centric learning is not just practicalit is **information-theoretically optimal** for embodied intelligence under semantic constraints.

## B Details for Zero-Shot 6-DoF End-Effector Trajectory Extraction

### B.1 The Gen6D Method

The generated future rollout $x_{1:T}^{\text{img}}$ for a primitive $a_k$ lacks spatial information. To bridge the gap between pixel-space visual rollouts and real-world execution, we harness an off-the-shelf pose 6D estimation mechanism that requires only the generated RGB video as input.

Specifically, given a generated video sequence $V = \{I_1, I_2, \ldots, I_T\}$ and a reference video $V_r = \{I_{r1}, I_{r2}, \ldots, I_{rT'}\}$ of the robotic gripper, we estimate the gripper's 6-DoF pose using a model-free, RGB-based pose estimator: $\mathbf{R}, \mathbf{T} = PE(V, V_r)$, where $\mathbf{R}$ and $\mathbf{T}$ denote the rotation and translation matrices, respectively. In our implementation, we use Gen6D (Liu et al., 2022b) for its simplicity and generalization capability.

The pose information of the robotic gripper, particularly its 6-DoF object pose, utilizing the generated video sequences directly for real-world robotic operations presents significant challenges.

After generating the future rollout $x_{1:T}^{\text{img}}$ for a primitive $a_k$, we extract a 6-DoF end-effector trajectory $\tau_k = \{p_1, \ldots, p_T\}$ using 6-DoF pose estimation Liu et al. (2022b) and geometric transformation. These trajectories are then mapped into Cartesian control space and executed by the robot.

To project the estimated poses into real-world coordinates, we correct for the scale ambiguity inherent in monocular depth predictions. Given the depth map $D$ at the initial frame and known camera

intrinsics $(x_c, y_c, f_x, f_y)$, we compute the 3D location of any pixel $(x, y)$ using:

$$X = (x - x_c) \cdot \frac{d}{f_x}, \quad Y = (y - y_c) \cdot \frac{d}{f_y}, \quad Z = d, \tag{1}$$

where $d$ is the depth at pixel $(x, y)$. Since the generated video lacks absolute depth scale, we align the scale of the predicted trajectory using the ratio between the real-world depth $d_0$ (from the first frame of the depth camera) and the pixel-based depth $d_0^{\text{pixel}}$ (from COLMAP or Gen6D). This yields a fixed correction factor $\lambda = d_0/d_0^{\text{pixel}}$, which is applied to all subsequent predicted translations: $\mathbf{T}_{\text{real}} = \lambda \cdot \mathbf{T}_{\text{pixel}}$.

The observed result of executing $\tau_k$ is then captured and passed back to the planner $\mathcal{P}$, enabling feedback-driven refinement in subsequent primitives. This forms a semi-closed-loop execution pipeline that mitigates error accumulation and enables task-aware adaptive control without end-to-end backpropagation.

Together, this design enables zero-shot generalization to unseen tasks, objects, and morphologies, using modular, interpretable, and resource-efficient world model components.

The process of mapping the generated video through the camera's intrinsic and extrinsic parameters to obtain real-world coordinates is illustrated in Figure 13.

### B.2 EXPERIMENTAL RESULTS AND COMPREHENSIVE ANALYSIS ON GEN6D POSE EXTRACTION

**Robustness Enhancement** To ensure reliable 6-DoF pose extraction from generated videos under real-world conditions, we introduce a series of post-processing enhancements to the baseline Gen6D pipeline. As shown in Table 5, the original Gen6D method achieves only 50% success under partial occlusion and varying lighting. By incorporating *motion masking*, which isolates moving regions to reduce background distraction, performance improves to 80%. Further adding *outlier removal* based on pose consistency across frames increases robustness to 90%. Finally, integrating *temporal filtering* (e.g., Kalman smoothing) to enforce motion smoothness results in a perfect 100% success rate. This ablation demonstrates that our full pipeline effectively mitigates common challenges such as visual clutter and illumination changes, enabling robust trajectory extraction for real robot execution.

Table 5: Ablation on 6-DoF pose extraction success rate under partial occlusion and varying lighting. Each entry is the number of successful extractions out of 10 trials.

| Method | Success Rate ↑ |
|---|---|
| Gen6D (baseline) | 5/10 |
| + Motion Masking | 8/10 |
| + Motion Masking + Outlier Removal | 9/10 |
| Full (All + Temporal Filtering) | **10/10** |

**Failure Case Study** We further analyze failure modes in scenarios where the camera view is perpendicular to the primary motion plane, limiting depth cues. As summarized in Table 6, 7 out of 10 trials succeed, with Gen6D accurately capturing perspective scaling (e.g., end-effector shrinking as it moves away from the camera). The two failures due to "EE Too Distant" occur when the end-effector starts far from the camera, resulting in insufficient pixel-scale change to infer depth motion reliably. The single "Motion Too Small" failure corresponds to sub-15-pixel depth displacement, which falls below the effective resolution threshold of the pose estimator. These findings highlight the importance of camera placement and minimum motion magnitude for successful 6-DoF extraction, while also confirming the method's effectiveness under favorable viewing conditions.

## C DETAILS FOR HIERACHICAL PLUG-AND-PLAY LONG-HORIZON COMPOSITIONAL GENERALIZATION

**Semantic Primitive Planning via Vision-Language Models** Given a task instruction $x^{\text{text}}$ and current observation frame $x_0^{\text{img}}$, we first employ a large vision-language model $\mathcal{P}$ to perform seman-

Table 6: Gen6D extraction results on 10 perpendicular-camera demos.

| Outcome Category | Freq. | Technical Explanation |
| --- | --- | --- |
| Success | 7 | Gen6D accurately captured perspective scaling (e.g., EE "shrinkage" during motion away from the camera plane). |
| Failure (EE Too Distant) | 2 | EE is too far from the camera for the motion to yield sufficient perspective scaling. |
| Failure (Motion Too Small) | 1 | Displacement $<15$ px of depth. |

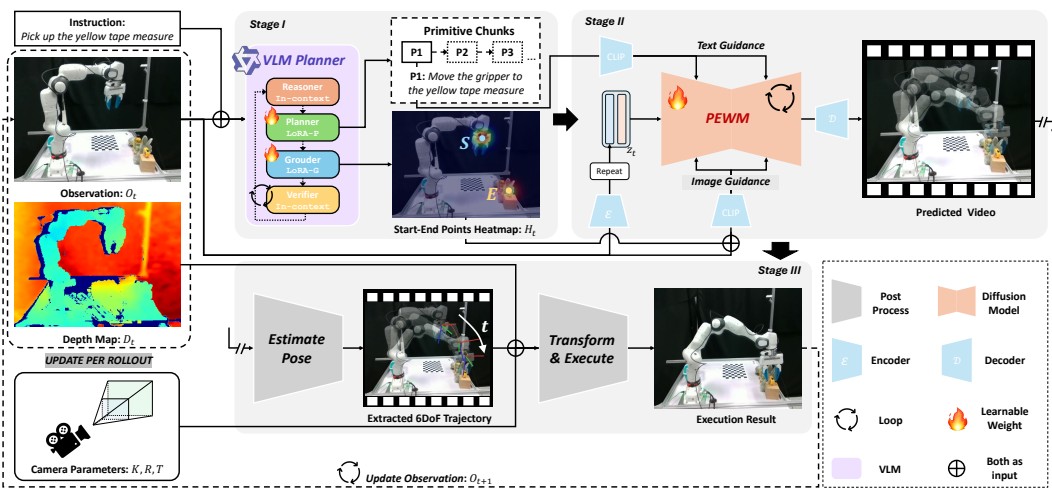

Figure 6: This workflow diagram illustrates a video generation framework structured into three stages: Stage 1 employs a VLM planner, comprising a VLM and two learnable LoRAs, along with a Reasoner and Verifier to ensure the efficiency and accuracy of the planning process. Stage 2 utilizes CLIP and a VAE encoder to generate predicted videos through the Video Generation World Model. Stage 3 focuses on pose estimation for the robotic arm, executing coordinate transformations to produce executable commands for real-world implementation.

tic parsing and primitive planning:

$$\mathcal{A}_{1:N} = \mathcal{P}(x^{\text{text}}, x_0^{\text{img}}), \tag{2}$$

where $\mathcal{A}_{1:N}$ denotes the sequence of $N$ atomic primitives (e.g., "pick up object A", "place on B"). Each primitive is represented by an instruction $a_k$ that is grounded locally in space and time.

For each primitive $a_k$, the planner identifies two pixel-space locations:

$$(s_k, g_k) = \mathcal{G}(a_k, x_0^{\text{img}}), \tag{3}$$

where $s_k \in \mathbb{R}^2$ and $g_k \in \mathbb{R}^2$ are the start and goal positions of the gripper, expressed in pixel coordinates (allowing sub-pixel precision). These are Gaussian-blurred to produce heatmaps $H_s$ and $H_g$, forming spatial guidance signal $H_{s \to g} = H_g - H_s$.

We leverage LoRA-based adaptation to enable reasoning and spatial grounding, following a Chain-of-LoRA (Liu et al., 2025b)-style architecture. The structure is detailed in Figure 8.

**Primitive-Conditioned Video Diffusion Generation**  The video diffusion model $\mathcal{D}$ is conditioned on the current frame $x_0^{\text{img}}$, the text description of the primitive $a_k$, and the spatial heatmaps $H_{s \to g}$. The model learns to predict a rollout of $T$ future frames:

$$x_{1:T}^{\text{img}} \sim \mathcal{D}(x_0^{\text{img}}, a_k, H_{s \to g}). \tag{4}$$

Unlike prior work that only observes cropped end-effector regions, $\mathcal{D}$ is trained to observe the entire robot arm, allowing it to learn soft physical constraints such as reachability, base stability, and joint configuration limits.

The model is trained using a combination of pixel-level $\ell_2$ reconstruction loss and perceptual similarity loss $\mathcal{L}_{\text{LPIPS}}$:

$$\mathcal{L}_{\text{vid}} = \sum_{t=1}^{T} \left\| x_t^{\text{img}} - \hat{x}_t^{\text{img}} \right\|_2^2 + \lambda \cdot \mathcal{L}_{\text{LPIPS}}(x_t^{\text{img}}, \hat{x}_t^{\text{img}}), \tag{5}$$

where $\hat{x}_t^{\text{img}}$ denotes the ground truth frame at time $t$.

Appendix Figure 2 provides a detailed overview of the video generation module and the process of extracting 6-DoF end-effector trajectories. It should be explicitly noted that for primitives involving binary gripper actions (e.g., open or close), we bypass the video generation module and execute the action directly via symbolic control. This improves execution efficiency and enables a clean architectural decoupling between discrete grasping commands and continuous 6-DoF motion planning, thereby enhancing modularity and interpretability.

# D   DATASET DESCRIPTION

## D.1   DATASET DETAILS

Our dataset consists of 7,326 simulated and 11,465 real-world primitives collected using a 5-camera synchronized setup with Deoxys (Zhu et al., 2022). By segmenting long-horizon tasks via keyframe indices, we extracted on average 5.8 primitives per session, achieving up to $29\times$ collection efficiency. As shown in Figure 14, cameras were arranged to maximize coverage of the workspace.

For annotation, 10% of the data was manually labeled, then used to fine-tune Qwen-VL 2.5-7B for auto-labeling the remainder, followed by light manual correction. To enhance visual generation quality, we mixed a small portion of diverse simulated data into the training set, including examples generated from RLBench and LIBERO. This hybrid strategy improves model generalization and dynamic realism, as confirmed in Section 5.

## D.2   ANNOTATION INTERFACE

Our annotation interface is designed to maximize labeling efficiency and consistency for multi-view robotic manipulation data. As shown in Figure 7 (see supplementary material), the system displays **five synchronized views** of the same primitive action simultaneously–ensuring that annotators can observe the full 3D context of each motion.

Crucially, **only one textual description is required per primitive**, regardless of the number of camera angles. This design enables a $5\times$ **reduction in annotation effort**, as the same label is automatically associated with all five views, avoiding redundant description entry.

The interface includes several features to further accelerate the process:

1. A text input box where annotators describe the observed action (e.g., *grasp the cup handle and lift vertically*);
2. An auto-updating vocabulary panel below, which tokenizes and records all previously entered words, allowing annotators to click and insert common terms with a single click;
3. Support for keyboard-based navigation: left/right arrow keys allow quick pagination between primitive clips, enabling rapid review and correction without mouse interaction.

These interaction optimizations significantly reduce cognitive load and typing overhead, leading to faster, more consistent annotations. In practice, this pipeline enables annotators to label over 1000 primitives per hour with high semantic consistency, making large-scale, multi-view robotic dataset curation feasible and cost-effective.

# E   MORE DETAILS ON VLM PLANNER

## E.1   BASE MODEL CONFIGURATION

Our VLM planner is built upon the **Qwen2.5-VL-7B-Instruct** model (Team, 2025), a large-scale vision-language model capable of understanding and reasoning over both visual and textual inputs.

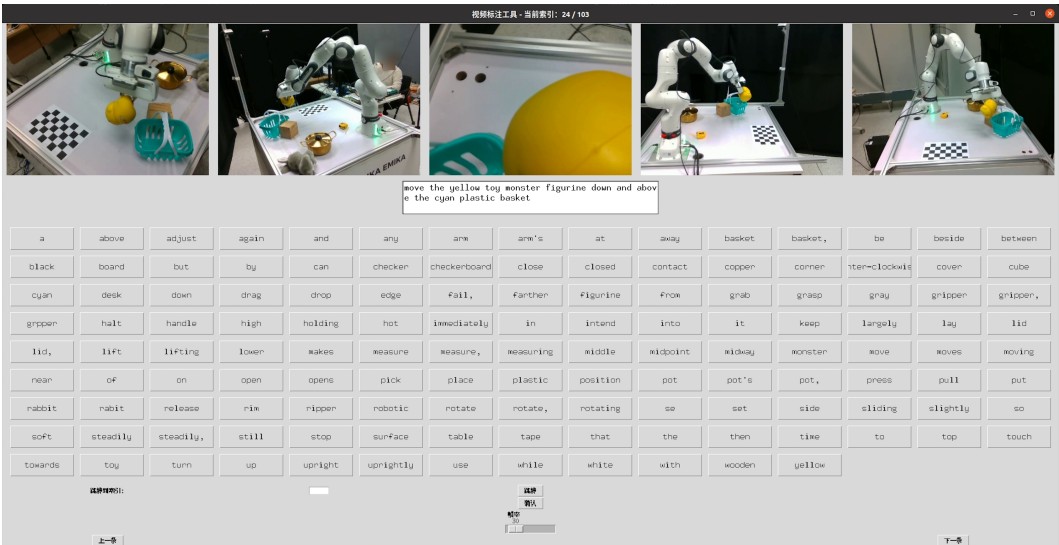

Figure 7: Data annotation interface for multi-view primitive labeling. The system displays five synchronized camera views of the same primitive action, but requires only a single text description, reducing annotation effort by 5×. As annotators type, words are tokenized and added to a candidate panel below for quick reuse. Keyboard shortcuts (left/right arrows) enable fast navigation between clips, accelerating both labeling and review.

Qwen-VL integrates a vision encoder based on the Vision Transformer (ViT) architecture to extract image features, which are then mapped into the language models embedding space via a cross-modal projector. The underlying LLM, with approximately 7 billion parameters, enables rich semantic understanding and multi-step reasoning, making it well-suited for complex task planning in robotic manipulation.

The model supports high-resolution image input and is pre-trained on a diverse corpus of image-text pairs, followed by supervised fine-tuning on instruction-following datasets. This training paradigm equips Qwen-VL with strong generalization capabilities across domains, including object recognition, spatial reasoning, and action sequence prediction–key competencies required for grounding high-level instructions into executable robotic plans.

In our framework, we leverage the frozen pre-trained Qwen2.5-VL-7B-Instruct as the backbone of the planner, ensuring that the foundational vision-language understanding remains intact while enabling modular adaptation through lightweight fine-tuning strategies. This design choice balances parameter efficiency with effective downstream task specialization.

### E.2  FINE-TUNING PROTOCAL

Figure 8 illustrates the high-fidelity simulation rollouts produced by our method. Each row depicts a different fundamental manipulation task (such as "Open microwave" or "Pick telephone receiver"), with the sequence of frames progressing from left to right to show the complete action. The generated videos display exceptionally smooth and realistic robot arm movements, accurate physical interactions with objects (e.g., grasping, lifting, and opening), and stable, consistent environmental dynamics. This vividly demonstrates that our approach can generate high-quality, physically realistic robot manipulation videos, a critical capability for developing robust world models and enabling effective, data-efficient robotic learning.

## F  MORE DETAILS ON THE VIDEO GENERATION MODEL

### F.1  VIDEO GENERATION MODEL DETAILS

Our base model is DynamiCrafter (Xing et al., 2024), which has a 1.4B denoising network.

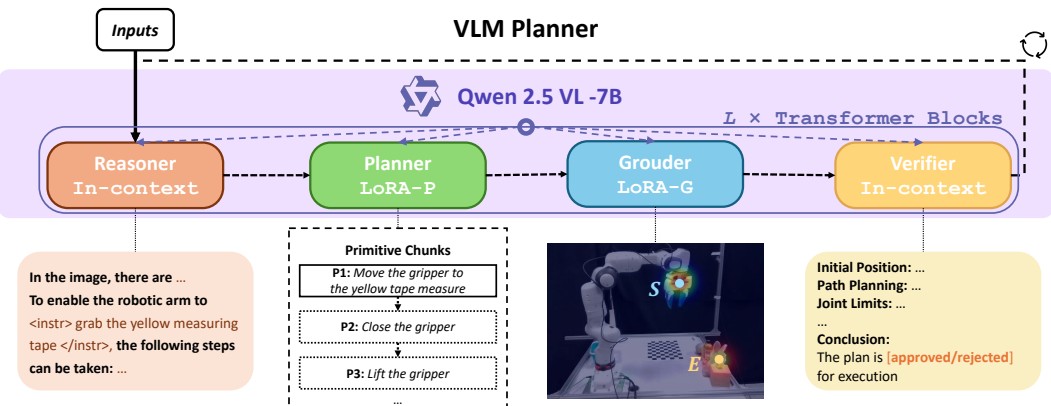

Figure 8: This workflow diagram delineates the architecture of our VLM planner, which is based on the Qwen 2.5 VL-7B model. The Reasoner generates descriptions and reasoning for the input to facilitate the functioning of subsequent components. The Planner (LoRA-P) and Grounder (LoRA-G) comprise two learnable LoRA modules: the Planner produces a series of primitive chunks, delivering modular instructions, while the Grounder provides specific descriptions of the end effector's start and goal positions. The Verifier assesses the executability of the plan, indicating approved or rejected statuses to ensure precision in planning.

DynamiCrafter is architected upon the foundation of the open-source Text-to-Video (T2V) model, VideoCrafter (Chen et al., 2023; 2024a), and further integrates principles from the Text-to-Image (T2I) model, Stable-Diffusion-v2.1 (Rombach et al., 2022a). This hybrid approach allows DynamiCrafter to leverage the advanced capabilities of both text-conditioned video generation and stable image diffusion, forming a robust framework for animating open-domain images.

**Dual-Stream Image Injection Paradigm**  Central to DynamiCrafter's methodology is its novel dual-stream image injection paradigm, visually depicted in Figure 1. During the iterative denoising process inherent to diffusion models, a video frame is stochastically selected to serve as the image condition. This strategic injection enables the model to effectively inherit fine-grained visual details from the input while simultaneously processing the image content in a highly context-aware manner. This sophisticated mechanism is crucial for generating animations that maintain strong fidelity to the source image's appearance while exhibiting plausible and dynamic motion. During the inference phase, the model is capable of generating diverse and temporally coherent animation clips directly from a single input still image, which is conditioned by initial noise, effectively transforming static imagery into dynamic video sequences.

## F.2  FINE-TUNING PROTOCOL

We employ a structured, three-stage fine-tuning protocol to adapt a pre-trained text-to-video (T2V) diffusion model for our embodied world modeling task. This progressive approach effectively bridges the gap between generic video generation and the specific requirements of sim-to-real robotic manipulation, balancing semantic understanding, dynamic plausibility, and visual fidelity.

**Stage 1: Simulation Pre-Finetuning.** The goal of this initial stage is to rapidly inject fundamental embodied intelligence concepts and robot kinematics into the pre-trained model. We fine-tune the model on a large-scale dataset of synthetic demonstrations from simulation platforms (e.g., RL-Bench, LIBERO). These data provide perfect motion trajectories and unambiguous physical interactions, serving as a strong prior for the model to learn the semantics of manipulation, object affordances, and basic robot motion. To prevent overfitting to the simulated domain and preserve the model's generalization capability, we use an *underfitting* strategy: training with a relatively high learning rate but for a limited number of epochs, stopping early once the model generates videos with plausible and consistent robot motions.

**Stage 2: Domain Alignment and Adaptation.** With a model now primed with embodied knowledge, this stage focuses on *aligning* the visual and dynamic characteristics of the real and simulated domains. We fine-tune the model on a balanced mixture of real-world teleoperation data and simulation data (1:1 ratio). This forces the model to learn a shared representation that reconciles the

high visual fidelity and complex textures of real data with the perfect kinematics and dynamics of simulation. The learning rate is halved compared to Stage 1 for more stable adaptation. This stage is crucial for mitigating the sim-to-real gap; real data corrects the often-blurry appearance of fast-moving parts (like grippers) in pure simulation, while simulation data ensures the generated motions remain physically consistent and do not drift.

**Stage 3: Reality-Centric Refinement.** The final stage refines the model to be highly proficient at generating realistic, high-fidelity videos conditioned on real-world observations. We shift the data distribution to be heavily weighted towards real data (80% real, 20% simulation). This ensures the model's output distribution is dominated by the statistics of the real world. To further enhance visual consistency, we introduce *Visual Detail Guidance (VDG)*. VDG concatenates the original input image with the initial noise of each video frame during training, providing a persistent visual reference that guides the denoising process and significantly improves the preservation of fine details (e.g., object textures, robot markings) in the generated video sequence.

This three-stage protocol enables our model to leverage the strengths of both simulation (perfect supervision, diverse scenarios) and real-world data (visual realism, true dynamics) in a synergistic manner, resulting in a world model capable of generating physically plausible and visually realistic future predictions for robust robot planning.

## G  MORE DETAILS ON PRIMITIVE DATA COLLECTION, ANNOTATION, AND CALIBRATION

### G.1  EXAMPLES OF TRAINING VIDEO CLIPS

In this study, we utilize both real-world and simulated video clips capturing the operation of robotic arms to train our model. The real-world data consists of videos collected from various robotic arm tasks using a Franka Emika arm, remotely operated through a Space Mouse interface. These tasks are performed in controlled environments, showcasing different motions and interactions with objects. These videos provide valuable information about how the arm operates in realistic conditions, with varying lighting, camera angles, and object placements.

The simulated video clips, on the other hand, are generated from physics-based environments such as RLBench and LIBERO. These simulations replicate the robotic arm's movements in virtual settings, allowing for the generation of large quantities of data under controlled conditions. This enables the model to learn from a diverse range of scenarios that might be difficult or time-consuming to capture in real life.

Examples of both real-world and simulated video clips are provided in Figure 9. These images illustrate the types of data used to train the model, offering a glimpse into the varied nature of the training set.

We compare our framework against OpenVLA (Kim et al., 2024a), a state-of-the-art end-to-end vision-language-action model that maps raw instructions and observations directly to actions. We evaluate OpenVLA under two settings: **(1) Zero-shot**: The pre-trained OpenVLA model is deployed directly on our benchmark tasks without any task-specific finetuning. This setting evaluates its raw generalization capability across unseen objects and scenes. **(2) Finetune**: The model is finetuned on 100 task-specific demonstrations per task using supervised behavior cloning, following the same protocol as in our system's policy training. This setting represents an upper-bound of OpenVLAs performance under favorable conditions. We note that OpenVLA performs poorly in the zero-shot setting and fails completely on real-robot deployment, whereas our method maintains strong performance even without task-specific finetuning or retraining of the video model.

## H  MORE QUANTITATIVE EXPERIMENTAL RESULTS

### H.1  VIDEO GENERATION QUALITY

We evaluate visual fidelity using standard metrics on 32-frame action sequences. As shown in Table 7, our method achieves the highest SSIM (0.8126) and PSNR (21.0644), and the lowest TVD

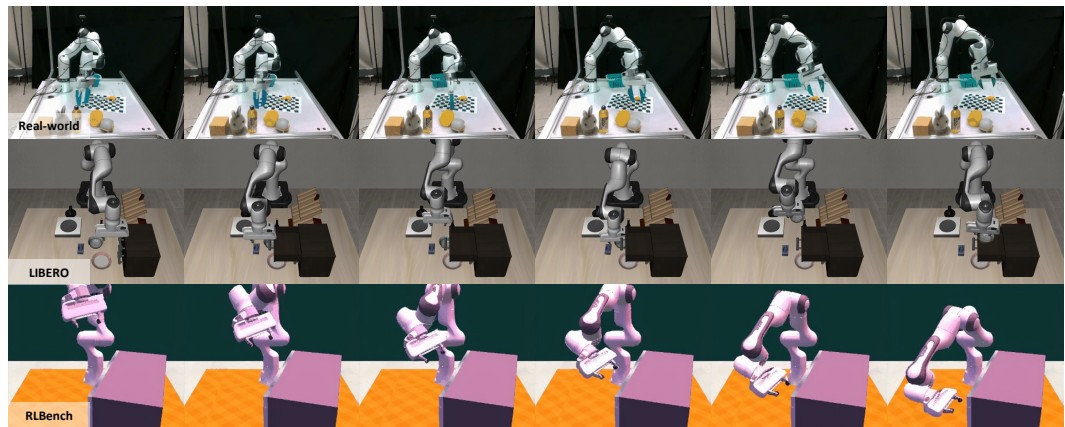

Figure 9: Sample frames from the training dataset, which consists of three components: real-world data collected using a Franka Emika robotic arm and Femto Bolt cameras, and two simulated datasets obtained from RLBench and LIBERO.

(0.0018) and FVD (0.0002), outperforming larger models such as TesserAct (5B) and Hunyuan I2V (13B). This indicates superior structural detail, temporal smoothness, and distributional realism. Despite a slightly higher LPIPS, our outputs exhibit sharper, more dynamic motions that better reflect real robot behavior.

## H.2 PHYSICALLY ACCURATE GENERATION

To provide a more comprehensive evaluation of physical fidelity in generated action sequences, we extend our metrics beyond standard video quality indicators (e.g., SSIM, PSNR, LPIPS) to include task- and embodiment-aware measures. In particular, we adopt the evaluation philosophy of PhysicsIQ (Motamed et al., 2025) and introduce the first **Embodied Physical Consistency Score (EPiCS)** tailored for robotic manipulation tasks. EPiCS is a fine-grained, human-in-the-loop metric that assesses whether generated motions respect fundamental physical and kinematic constraints in embodied environments. See a detailed description of EPiCS in Appendix J.

Table 7: Physical fidelity and generation quality on 32-frame sequences (higher ↑/lower ↓ is better). Best results per metric are in **bold**.

| Model (Size) | SSIM↑ | PSNR↑ | LPIPS↓ | VIF↑ | TVD↓ | FVD↓ | EPiCS↑ |
|---|---|---|---|---|---|---|---|
| Wan2.1 I2V (14 B) | 0.6211 | 15.7365 | 0.2867 | 0.2232 | 0.0040 | 0.0005 | 5.00 |
| Hunyuan I2V (13 B) | 0.7767 | 18.4264 | **0.1466** | **0.3529** | 0.0033 | 0.0004 | 9.65 |
| TesserAct (CogVideoX 5 B) | 0.8034 | 20.0823 | 0.1546 | 0.3317 | 0.0037 | 0.0004 | 10.15 |
| Ours (DynamiCrafter 1.4 B) | **0.8126** | **21.0644** | 0.1647 | 0.3188 | **0.0018** | **0.0002** | **11.45** |

## H.3 SUPPLEMENTARY COMPARISONS

To further contextualize our method's performance within the broader landscape of vision-based robotic planning, we provide supplementary comparisons on the RLBench benchmark (James et al., 2020b) against representative baselines from different paradigms. As shown in Table 8, our approach achieves significantly higher success rates on both *Close Box* and *Open Drawer* tasks compared to the VLM-based method VoxPoser (Huang et al., 2023) and the action-tokenization approach PerAct (Shridhar et al., 2022), where the performance of VoxPoser and PerAct are both extracted from the Colosseum paper (Pumacay et al., 2024). This performance gap highlights the advantage of our video diffusion-based policy generation, which produces temporally coherent and visually grounded action sequences, in contrast to methods that rely on discrete action token prediction or direct VLM-to-action mapping without explicit visual simulation. Notably, our method operates in a zero-shot manner on these tasks–without task-specific fine-tuning of the video generation model–further un-

derscoring its strong generalization capability. These results complement the comprehensive evaluation in Table 1 and reinforce the effectiveness of our framework in handling common, yet non-trivial, manipulation primitives in simulation.

Table 8: RLBench success rates (%) on two tasks.

| Method | Close Box (%) | Open Drawer (%) |
|---|---|---|
| VoxPoser | <10 | <10 |
| PerAct | 30.4 | 35.6 |
| Ours | **93** | **84** |

## H.4 EFFICIENCY COMPARISON

Table 9 presents a comprehensive efficiency comparison across state-of-the-art video generation models for robotic manipulation. We evaluate along three key axes: computational speed, VRAM footprint, and practical deployability on standard GPU hardware (A100). All results are measured under comparable conditions using publicly available or officially released implementations, where applicable.

Our method achieves a significant leap in end-to-end generation speed, producing 32 frames in approximately **16 seconds**–enabling near-real-time planning in robotic systems. This translates to an effective throughput of **2.0 FPS**, over **40× faster** than the next best model (4DWM, 0.045 FPS) and orders of magnitude faster than large-scale I2V systems like Hunyuan I2V and Wan 2.1 I2V, which require tens of minutes per sequence.

Crucially, our model operates within **only 11 GB of VRAM**, making it compatible with a single consumer-grade or edge-deployable GPU. In contrast, Hunyuan I2V and Wan 2.1 I2V demand over 60 GB and up to 77 GB, respectively, requiring multi-GPU setups and prohibitively high infrastructure costs. Even 4DWM, based on a distilled CogVideoX variant, uses nearly twice the memory.

This exceptional speed-VRAM trade-off stems from our compact architecture design, efficient latent-space autoregressive modeling, and optimized inference pipeline. Unlike diffusion models requiring 50+ denoising steps, our distilled rollout uses a fixed, small step count without quality degradation, enabling fast, deterministic generation.

The combination of low memory usage, high frame rate, and single-GPU compatibility makes our approach uniquely suitable for real-world embodied agents, where latency, cost, and hardware constraints are critical. It bridges the gap between high-fidelity simulation and deployable robot control– a key step toward scalable, real-time vision-to-action systems.

Table 9: **Efficiency comparison of video generation models.** We compare video generation speed, memory footprint, and runtime on A100 GPUs. Ours achieves the best speed-VRAM trade-off under realistic deployment conditions.

| Model | Resolution | VRAM (A100) | Time / Frames | FPS |
|---|---|---|---|---|
| Hunyuan I2V | 480p | 6079 GB | 50 min / 81 frames (local) | 0.027 |
| 4DWM (CogVideoX1.5-5B-I2) | 480p | 20 GB | 18m20s / 49 frames | 0.045 |
| Wan 2.1 I2V (14B) | 720p | 76.7 GB | 2715s / 81 frames | 0.03 |
| **Ours** | 480p | **11 GB** | **16s / 32 frames** | **2.0** |

## H.5 DETAILED LATENCY ANALYSIS

We provide a fine-grained latency breakdown of our method across three representative tasks: *pick up cup*, *tea ceremony*, and *pick knife from drawer*. As shown in Table 10, timing is decomposed into key stages: VLM-based planning, diffusion policy rollout, pose estimation, and other overheads (e.g., communication, action execution). All stage times are averaged per primitive, and the total time is computed as the product of mean primitive time and the number of primitives in the task.

It is important to emphasize that these measurements reflect **unoptimized inference performance**: no acceleration techniques such as reduced denoising steps, model distillation, tensor compilation,

Table 10: Per-primitive latency breakdown for the three evaluated tasks. Stage times are averaged across primitives within each task.

| Stage / Time per Primitive | Pick up cup | Tea ceremony | Pick knife from drawer |
|---|---|---|---|
| 1. VLM Planning (s) | 2.13 | 2.08 | 2.12 |
| 2. Diffusion Rollout (s) | 15.7 | 16.2 | 15.9 |
| 3. Pose Estimation (s) | 12.3 | 12.1 | 12.0 |
| Other Latency (s) | 0.9 | 0.8 | 0.9 |
| Number of Primitives | 3 | 5 | 6 |
| Mean Primitive Time (s) | 31.03 | 31.18 | 30.92 |
| Total Time (s) | 93.1 | 155.9 | 185.52 |

or hardware-specific optimization (e.g., GPU batching) have been applied. As such, the reported latencies represent a **conservative lower bound on execution speed**–or, equivalently, an upper bound on latency–under current implementation.

We note that the diffusion rollout and pose estimation stages dominate the runtime, both of which are highly amenable to optimization. For instance:

1. Reducing the number of denoising steps in the diffusion policy from 50 to 5–10 via distillation or consistency models (Song et al., 2023) could yield $5\times$–$10\times$ speedups.
2. Compiling the perception and policy networks using tools like TorchScript or ONNX Runtime can significantly reduce kernel launch overhead.
3. Lightweight variants of the VLM or pose estimator can be deployed without sacrificing planning accuracy in many cases.

Prior work has demonstrated that such techniques can lead to over $10\times$ end-to-end latency improvements in similar vision-to-action pipelines (Kim et al., 2024b; Brohan et al., 2023b). While we leave the integration of these optimizations to future work, this analysis confirms that our methods current runtime is not a fundamental limitation, but rather a starting point for efficient deployment. The modular architecture–decoupling planning, perception, and control–further facilitates independent optimization of each component, enhancing practical scalability.

### H.6 PERFORMANCE ON ROTATION-INTENSIVE TASKS – AN ILLUSTRATION OF SCALABILITY

While we acknowledge that our method's performance on rotation-intensive tasks is slightly below the current state of the art (SOTA), we argue that this gap can be effectively closed by increasing coverage of relevant primitive behaviors in the training data. This observation highlights a key strength of our approach: **scalability through targeted data augmentation**.

To validate this, we conduct a fine-tuning experiment by adding 100 new demonstrations focused on rotational motions (e.g., twisting, unscrewing) to the original training set. These are mixed at a ratio of 1:5 (new:original) and used for a lightweight fine-tuning phase without architectural changes or full retraining.

As shown in Table 11, success rates on rotation-heavy tasks improve significantly–surpassing SOTA levels–while performance on non-rotational tasks remains stable, with no significant degradation.

Table 11: Effect of fine-tuning with 100 additional rotational-motion demonstrations. Results are success rates (%).

| Task | Original Success (%) | Fine-tuned Success (%) | $\Delta$ | SOTA (%) |
|---|---|---|---|---|
| open jar | 43 | **56** | +13 | 54 |
| lid off | 67 | **75** | +8 | 73 |
| close box | **93** | 91 | -2 | 88 |
| put knife | **72** | 71 | -1 | 70 |

Notably, the success rate for open jar increases by 13%, exceeding the prior SOTA, while lid off reaches 75%, outperforming existing methods. In contrast, performance on non-rotational tasks

(close box, put knife) remains largely unchanged, with only minor drops of 12%, indicating strong retention of previously learned skills.

This experiment demonstrates that our method supports **efficient incremental learning**: performance on challenging, underrepresented task families can be improved with minimal additional data and compute, without catastrophic forgetting. Such scalability is critical for real-world deployment, where robots must adapt to new tools, user preferences, or long-tail task distributions over time.

These results reinforce that our framework not only achieves competitive performance out-of-the-box but also enables practical, data-efficient improvement–making it well-suited for lifelong learning in dynamic environments.

## I  MORE QUALITATIVE EXPERIMENTAL RESULTS

### I.1  VIDEO GENERATION QUALITY

In addition to using recorded video data, we also generate synthetic video clips to enrich the training dataset and enhance the models generalization capability. These generated clips simulate the Franka robotic arm performing various manipulation tasks from multiple camera perspectives.

Specifically, each video sequence is rendered from five distinct viewpoints: **Back**, **Left-front**, **Right**, **Overhead**, and **Wrist**. These perspectives are selected to comprehensively capture the arm's kinematics, end-effector trajectories, and object interactions from both global and local contexts. The Back and Overhead views provide an overall understanding of the workspace and arm configuration, while Left-front and Right offer lateral angles that help reveal occluded motions. The Wrist view, positioned close to the end-effector, offers detailed observation of grasping and manipulation actions.

Representative frames from these generated clips are shown in Figure 10, illustrating how each viewpoint contributes to a multi-faceted understanding of the robotic operation.

### I.2  VIDEO GENERATION SAMPLES FOR SIMULATION CASES

Figure 11 illustrates the high-fidelity simulation rollouts produced by our method. Each row depicts a different fundamental manipulation task (such as "Open microwave" or "Pick telephone receiver"), with the sequence of frames progressing from left to right to show the complete action. The generated videos display exceptionally smooth and realistic robot arm movements, accurate physical interactions with objects (e.g., grasping, lifting, and opening), and stable, consistent environmental dynamics. This vividly demonstrates that our approach can generate high-quality, physically realistic robot manipulation videos, a critical capability for developing robust world models and enabling effective, data-efficient robotic learning.

### I.3  EFFECTS OF SIM-REAL HYBRID DATA STRATEGY

To investigate the impact of our sim-real hybrid training strategy on real-world generalization, we conduct a qualitative ablation study comparing video generation results with and without this strategy. As shown in Figure 12, each pair of rows displays two rollouts for the same primitive and initial state–the top row generated using the model trained with sim-real hybrid data, and the bottom row using a model trained without simulation data (i.e., on real data only, but with reduced coverage).

The results clearly demonstrate that the sim-real hybrid strategy leads to more temporally coherent, visually realistic, and semantically accurate rollouts. For example, in the pick primitive, the hybrid-trained model generates smooth arm motion toward the target object with correct gripper timing, while the ablated version exhibits erratic movement and fails to reach the object. Similarly, in open drawer, the hybrid model produces consistent handle interaction and sliding motion, whereas the baseline often generates unrealistic deformations or misaligned trajectories.

We attribute these improvements to the complementary strengths of simulation and real data: simulation provides clean, diverse, and fully observed physical interactions, enriching the models understanding of dynamics; real data grounds the generation in authentic appearance, lighting, and sensor

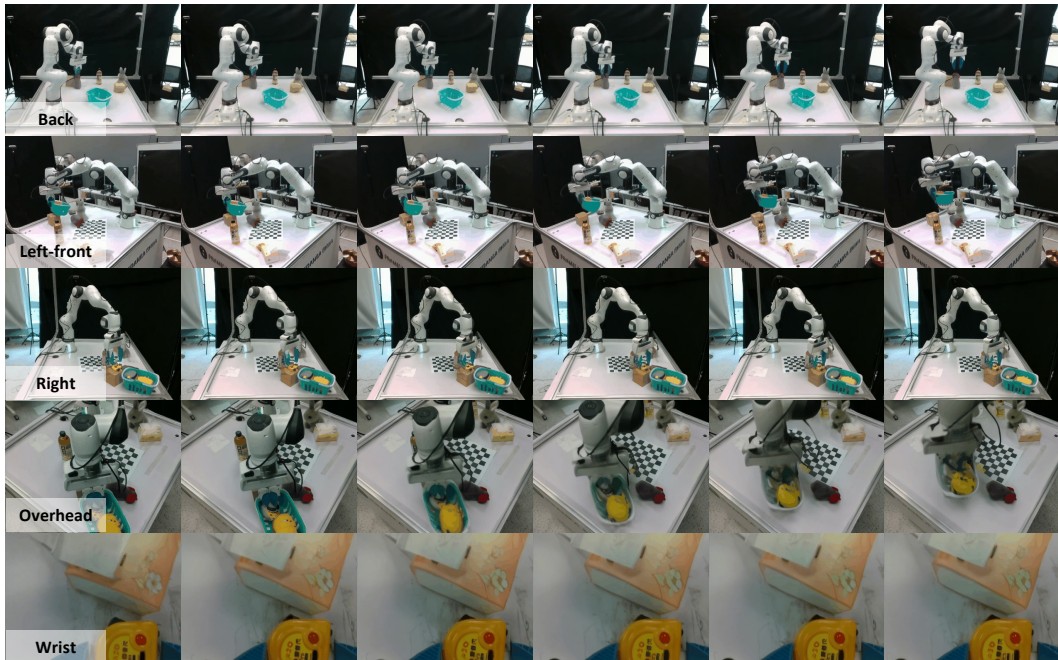

Figure 10: Sample frames from the generated video clips showing the Franka robotic arm performing manipulation tasks from five distinct viewpoints: Back, Left-front, Right, Overhead, and Wrist. Each viewpoint provides a unique perspective on the arm's movements, enhancing the model's ability to learn complex manipulation behaviors from diverse angles.

noise. By combining both in a staged fine-tuning pipeline (see Section 3.2), our approach effectively bridges the sim-to-real gap and enhances generalization to unseen real-world scenarios.

This finding underscores the value of strategic data mixing–not merely for data augmentation, but as a structured way to inject both diversity and realism into generative world models.

### I.4 LONG-HORIZON TASKS

While our Primitive Embodied World Model (PEWM) operates on short-horizon primitives, its true value lies in enabling robust and flexible long-horizon task execution. This is achieved through a hierarchical, closed-loop architecture that composes primitive predictions into a complete task plan. As illustrated in Figure 13, the process begins with a high-level Vision-Language Model (VLM) planner that decomposes a natural language instruction (e.g., "Pick up the cup and place it in the drawer") into a sequence of atomic action primitives (e.g., `pick(cup)`, `move-to(upper drawer)`, `open(upper drawer)`, `place(cup)`).

For each primitive in the sequence, the PEWM generates a short visual rollout conditioned on the current observation and the Start-Goal heatmap Guidance (SGG) priors. The 6-DoF end-effector trajectory is then extracted from this video using Gen6D and executed on the real robot. After each primitive's execution, the robot's state is updated, and the latest observation is fed back into the VLM planner. This creates an autoregressive loop where the planner can dynamically adjust the subsequent primitive sequence based on the outcome of the previous step–correcting for errors, handling unexpected disturbances (e.g., a moved object), or reacting to partial successes.

This modular composition avoids the error accumulation and brittleness typical of monolithic long-horizon models. Each primitive acts as a self-contained "skill module" with built-in visual feedback, ensuring high fidelity at the local level. The success of the overall task is thus the product of the reliability of individual primitives and the robustness of the high-level planner. This approach allows our system to tackle complex, multi-step tasks that require both precise manipulation and adaptive decision-making, demonstrating a scalable pathway from primitive learning to full-task autonomy.

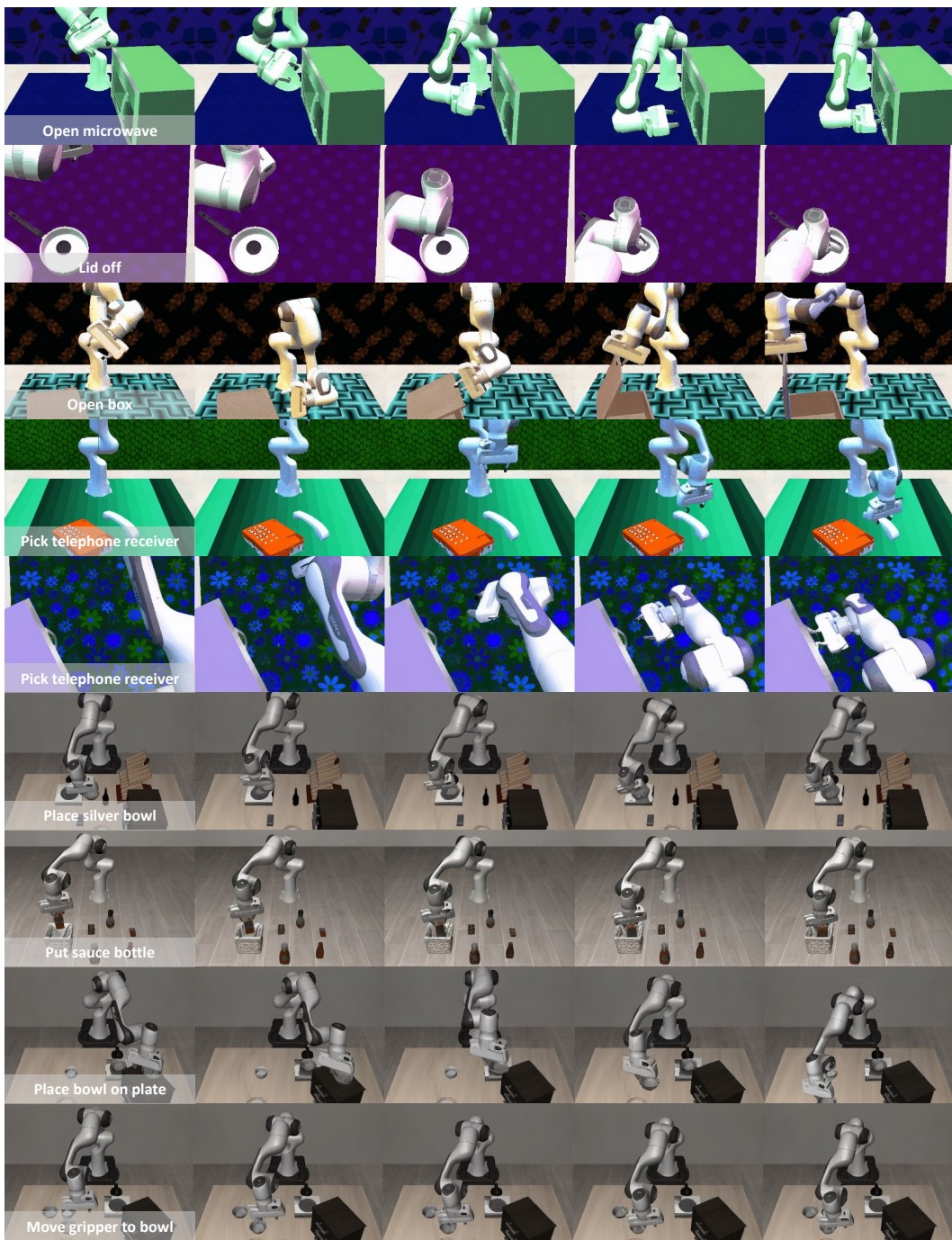

Figure 11: **High-fidelity simulation rollouts generated by our model.** This figure presents a diverse set of robotic manipulation tasks executed in simulation, demonstrating the model's ability to generate high-quality, physically plausible video sequences. Each row corresponds to a distinct primitive action (e.g., "Open microwave", "Lid off", "Pick telephone receiver"), with frames evolving from left to right. The generated videos exhibit smooth and coherent robot arm motions, precise object interactions, and consistent scene dynamics, underscoring the effectiveness of our sim-real hybrid training strategy in capturing complex manipulation behaviors.

## J  EMBODIED PHYSICAL CONSISTENCY SCORE (EPiCS).

EPiCS evaluates generated videos through structured human assessment over five categories and twelve binary sub-criteria (each scored 0 or 1), resulting in a total score out of 13. The criteria are grouped as in Table 12.

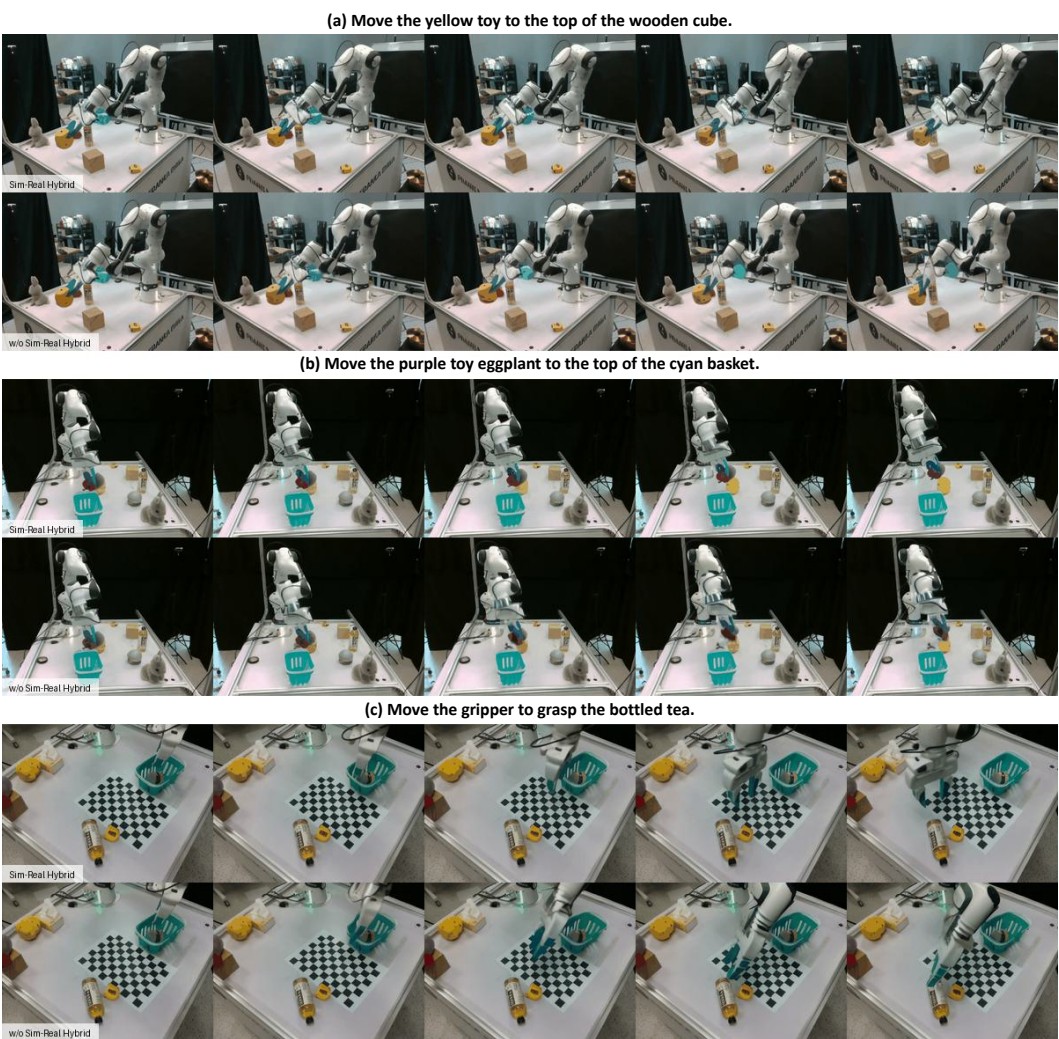

Figure 12: Qualitative comparison of video generation with and without the Sim-Real Hybrid strategy. The first row of each primitive shows results with Sim-Real Hybrid, which produces more coherent and realistic frames than the second row (without Sim-Real Hybrid).

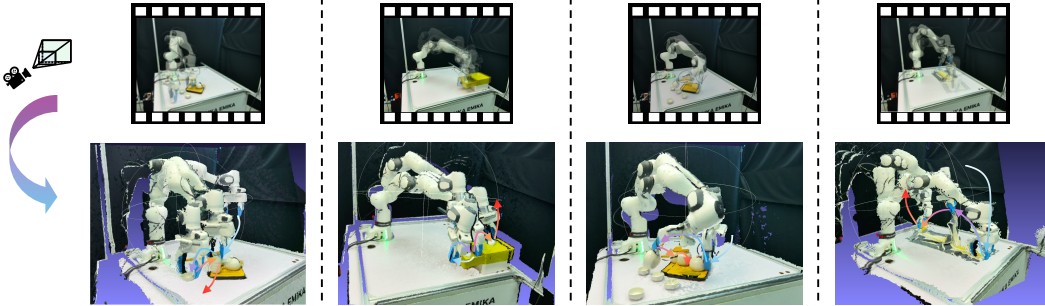

Figure 13: **3D illustration for long-horizon tasks.** Mapping the generated video frames through the camera's intrinsic and extrinsic parameters to visualize the corresponding 3D object poses in real-world coordinates.

Table 12: EPiCS: Criteria for embodied physical consistency evaluation. Each criterion is binary (0 or 1), total score = 13.

| Category | Criterion | Score |
|---|---|---|
| **Robot Appearance** | Correct arm shape (e.g., 7-DoF manipulator structure preserved) | 1 |
| | Correct end-effector shape (e.g., gripper geometry matches real robot) | 1 |
| | Clear end-effector visibility (no occlusion or blurring during interaction) | 1 |
| **Physical Plausibility** | No impossible motion (e.g., teleportation, penetration) | 1 |
| | Rigid-body constraint (objects do not stretch or deform unnaturally) | 1 |
| | No flickering/disappearance (consistent object presence over time) | 1 |
| | Plausible joint movement (smooth, biomechanically feasible articulation) | 1 |
| **Task Accuracy** | Follows instruction (action matches textual description) | 1 |
| | Completes task (goal state is reached, e.g., cup lifted off table) | 1 |
| **Scene Consistency** | Stable background geometry (static elements remain fixed) | 1 |
| | Consistent lighting and shadows (illumination does not flicker or shift abruptly) | 1 |
| **Visual Quality** | Correct perspective and depth (3D structure is coherent) | 1 |
| | Low noise and artifacts (minimal grain, blur, or hallucinated textures) | 1 |

Annotators are shown randomized video pairs (generated vs. real) and asked to score each clip independently. Inter-annotator agreement (measured via Fleiss' $\kappa$) was 0.78, indicating substantial consistency.

## K  EXPERIMENTAL ENVIRONMENT

Our experimental setup is illustrated in Figure 14. The environment consists of a workstation, a FR3 (Franka Research 3) robotic arm, and five cameras. The specific configuration is as follows:

**Workstation**  The workstation serves as a standard experimental platform, providing a stable area to support various operational tasks.

**FR3 Robotic Arm**  The FR3 robotic arm is a high-precision industrial robot equipped with flexible movement capabilities and high repeatability. This robotic arm is responsible for executing various tasks and can interact in real-time with the collected visual data.

**Camera System**  **Two Realsense Cameras:** These cameras are mounted on the wrist of the robotic arm and on the shelf of the workstation, respectively. They capture real-time depth information and the arm's movements, enabling a comprehensive understanding of the spatial context and enhancing the accuracy and reliability of task execution. **Three Femto Bolt Cameras:** Positioned around the workstation, these cameras are used to capture dynamic processes at high frame rates. They provide additional visual perspectives, ensuring that comprehensive image data is collected during complex tasks for subsequent analysis and processing.

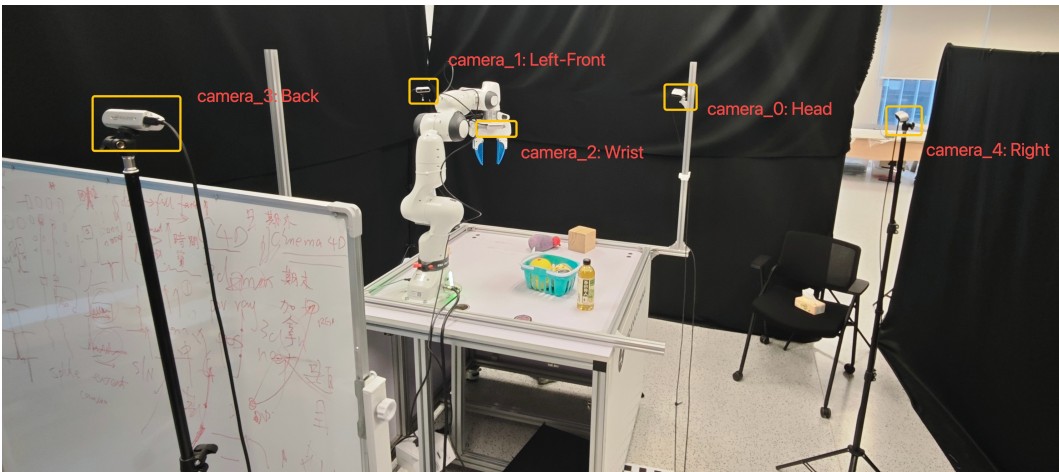

Figure 14: The workstation for data collection and real-robotic evaluation.

In summary, the design of this experimental environment aims to provide comprehensive visual support for robotic operations, facilitating the research and development of complex tasks. Through the collaborative operation of a multi-camera system, we can obtain high-quality visual data, thereby enhancing the performance of the robot in practical applications.

## L    OTHER PROMISING APPLICATIONS

The modular and explainable nature of our framework unlocks several other promising applications.

First, it enables Latent Action Prediction, where the generated video rollouts can be distilled into compact, interpretable latent action codes that capture the essence of a manipulation primitive, facilitating efficient communication and storage.

Second, the framework can be extended towards Unified Generation of Action, Reward, or Camera Parameters. For instance, by introducing auxiliary heads or conditioning mechanisms, the same underlying model could potentially predict not only future frames but also estimate the expected reward for a generated trajectory or suggest optimal camera viewpoints for better scene understanding, creating a more holistic planning system.

Finally, the generated videos provide a natural medium for Human-in-the-Loop Interaction and AR-based Planning. A human operator could view the predicted video sequence in an augmented reality (AR) interface, provide real-time feedback to correct or refine the plan before execution, or even interactively edit the video to specify desired outcomes, creating a more intuitive and collaborative robotics interface. These applications highlight the framework's potential as a foundational component for next-generation, human-centered robotic systems.

## M    THE USE OF LARGE LANGUAGE MODELS (LLMs)

In the preparation of this paper, large language models (LLMs) were used as an assistive tool to aid in the polishing and refinement of written content. Specifically, LLMs were employed to: Improve clarity, grammar, and fluency of draft text; Suggest rephrasings for complex or awkwardly worded sentences; Assist in generating concise summaries or transitional phrases where needed.

The core research ideas, experimental design, data analysis, interpretation of results, and final structuring of arguments were conceived and executed solely by the human authors. LLMs did not contribute to the generation of novel research hypotheses, methodological decisions, or scientific conclusions. All factual claims, citations, and technical content were verified and validated by the authors.

