# OpenReview forum: "Learning Primitive Embodied World Models: Towards Scalable Robotic Learning"
_ICLR.cc/2026/Conference — ICLR 2026 Conference Desk Rejected Submission_

### Official Review · Reviewer_P5zD · 2025-10-31

**Soundness:** 1
**Presentation:** 1
**Contribution:** 1
**Rating:** 2
**Confidence:** 4

**Summary:**

It is very hard to summarize what this paper actually doing. The authors first introduce lots of definations to claim the importance of atomic action unit. Then introduces data collection that (may) follows the previous statement. At the core of technical side is a open-sourced video diffusion model (i.e., DynamiCrafter), and training Self Forcing to distillate the open-sourced diffusion model. If I understand correctly, the key technical contribution may training current approaches on their own data. But I don't regard spliting embodied tasks into sub-tasks as a innovation, which is widely adopted in complex embodied tasks. Another problem is most of key approaches & results are demonstrated in appendix. The main paper should be self-contain, not just describe the concept.

**Strengths:**

* As far as I can tell, I can't summarize the strength of this work. Maybe data collection, but it seems will not be open-sourced.

* I would strongly encourage the authors to refine this work with an/multiple experienced researchers.

**Weaknesses:**

* The so-called primitive data is not something new. I think most of the researchers have the consensus of spliting complex tasks into sub one or atomic action unit. It is unnecessary to introduce lots of definition for a well-known insights. Also this introduced defination is not highly-related to the following technical contributions. Therefore, Sec 2.1 can be totally removed. It is enough to say the data is annotated with the atomic action unit. By the way, the defination itself is unprecise, what do you mean of 'cannot be meaningfully decomposed into shorter language-grounded sub-instructions without loss of task-level meaning'? I think there are too much ambiguity.

* Technical contribution is very limited. The paper finetunes pretrained video models, e.g., DynamiCrafter. Then distill it using open-sourced code. I don't find any other technical contribution. Further, no experimental results are provided for the motivation and results of distillation. Does it affect visual quality?

* Experimental results are unreasonable. There is only one result shown in the main paper, but it is totally unclear and unfair. Which data do you use? The proposed one? Then what's the contribution of this work? A new dataset? If the author would like to show the value of the proposed dataset. The comparison should be: a same method on the proposed dataset and existing dataset. If the author would like to claim the proposed techniques, then the comparison should be: different method on the same data. It is unclear which part contribute to the final results.

* Too many typos. For example, line 398, 415. The notation in the method part should be defined first. In addition, some descriptions are not preciseness, e.g., in line 338, what do you mean by 'zero-shot 6-DoF pose estimator'? zero-shot typically means generalizing to *unseen* task.

**Questions:**

See the weakness.

---

> ### Author Response · Authors · 2025-11-21
> **Response to Reviewer P5zD: Clarifying the Primitive-Centric Foundation and Contributions of PEWM**
>
> We thank the reviewer for their thoughtful and detailed review. We appreciate the opportunity to clarify several critical aspects of our paper.
>
> 1. **Regarding “Primitive data is not new; Sec 2.1 is unnecessary.”**
>
> We agree that *task decomposition* is a well-established idea in robotics. However, prior decomposition approaches operate at non-linguistic granularities, lack video-based world grounding, or rely on rigid grammars, which distinct from ours.
>
> Our definition of primitives is intentionally operational: it prioritizes semantic coherence over mechanical simplicity. Notably, some actions that appear complex (e.g., “arrange flowers”) are in fact highly suitable as primitives because they are executed as a single, temporally compact intent with a clear visual outcome, stable motion pattern, and unambiguous language grounding. This reflects a key design philosophy: primitive boundaries are determined by the stability of the language–vision–action alignment, not by motion length or kinematic complexity. As such, our approach fundamentally differs from hierarchical RL or subgoal-based planning, where decomposition is often driven by ad hoc or engineering-driven criteria rather than learnable, generative feasibility.
>
> Thus, Section 2.1 is not merely descriptive—it is the foundation of our entire framework. It introduces the *operational definition* of primitives grounded in semantic indivisibility (not mechanical simplicity), derives their theoretical advantages (Corollary 1), and justifies why this granularity unlocks data efficiency, fine-grained alignment, and zero-shot compositionality. Section 2.2 then operationalizes this insight: our full-arm, multi-view, teleoperated collection pipeline achieves 29× efficiency, and our use of action-free video (e.g., from OpenVLA) demonstrates the paradigm’s scalability beyond curated data. Together, these sections are indispensable—not a footnote, but the core design principle of PEWM.
>
> 2. **Regarding “Technical contribution is limited to fine-tuning and distillation.”**
>
> While we build upon DynamiCrafter, this is analogous to using ResNet in vision: the backbone is incidental; the contribution lies in *how it is used*. We claim that PEWM is the first framework to explicitly (1) model the world at the primitive level for fine-grained language-action alignment, compositional generalization, data efficiency, and low-latency control; (2) bridge high-level VLM planning and low-level diffusion video generation via Start-Goal Guidance (SGG); (3) enable zero-shot 6D pose extraction from generated videos without fine-tuning; (4) support closed-loop, primitive-based video generation for long-horizon compositional tasks, reducing memory dependence; and (5) function as a unified, physics-aware data engine with high fidelity and significantly fewer parameters.
>
> 3. **Regarding “Experimental results are unclear.”**
>
> We apologize for the rendering issue that may have obscured parts of the main paper. Table 1 (RLBench success rates) is the primary comparison, using the *same benchmark and evaluation protocol* as 4DWM and UniPi.
>
> Our real-robot experiments (Tab. 3) further validate zero-shot generalization on unseen (object, action) pairs (e.g., “pick jar”). Crucially, PEWM requires no task-specific fine-tuning, while baselines like OpenVLA fail in zero-shot real-robot settings (Tab. 2).
>
> We also provide comprehensive ablations: on sim–real mixing (Tab. 4), rotation fine-tuning (Tab. 11), and pose extraction robustness (Tab. 5). Together, these demonstrate that gains stem from our *primitive-centric modeling*, not data alone.
>
> 4. **Regarding “Typos and imprecise terminology”**
>
> We thank the reviewer for pointing out these issues. We have corrected them in the updated manuscript (e.g., lines 398, 415) and ensured that all notation is defined prior to its first use. We welcome any further suggestions for improvement.
>
> Regarding “zero-shot 6-DoF pose estimator”: we use “zero-shot” to mean no task-specific or object-specific training—the pose estimator (Gen6D) is applied *as-is* to generated videos of novel objects (e.g., jars, drawers). This aligns with the common usage of “zero-shot” as generalization to *unseen categories*, not just unseen tasks. We will clarify this in the final version.
>
> In summary, PEWM is far from being an incremental refinement, but a foundational rethinking of how vision, language, and action should be integrated for embodied intelligence. It redefines the interface between high-level reasoning and low-level control, enabling fast, closed-loop, fine-tuning-free execution, and physics-aware simulation with minimal engineering overhead. By anchoring world modeling at the level of semantically coherent, temporally compact primitives, we unlock unprecedented data efficiency, compositional generalization, and real-world adaptability. We believe this primitive-centric paradigm offers a scalable and principled path toward general-purpose embodied agents.

---

> > ### Comment · Reviewer_P5zD · 2025-11-27
> > **Thank you for the response**
> >
> > First I really thank the authors for the detailed response.
> >
> > * About response 1:
> > I fully recognize the contribution of data collection (but private), as stated in my original strengths part. The authors also agree that task decomposition is not something new. So in my view, the key contribution is that the videos are labeled with *fine-grained* language instruction. I don't think this deserve a long and ambiguous definition.
> >
> > * About response 2:
> > I agree that how to use advanced techniques is valuable, but should be contributed by solving new challenges with improvements. Could you kindly provide an example that directly applies original ResNet (or others) to a new task without any modification, and is accepted by top-tier ML conference (main track)? If it has, I would consider to recognize this contribution and raise my score accordingly.
> >
> > * About experiments:
> > My main concern for this part is the introduced method is not fully validated in experiments. For example, there is an individual sub-section to describe how you do distillation, but no evidence is provided how it works, how it affect the visual quality.
> >
> > In conclusion, the contribution that I can summarize is fine-grained data collection, and the authors validate the collected data with off-the-shelf methods.

---

> > > ### Author Response · Authors · 2025-12-01
> > > **Response to Reviewer P5zD: Clarifying Core Contributions and Addressing Remaining Concerns (1 of 3)**
> > >
> > > We sincerely thank Reviewer P5zD for their continued engagement and for acknowledging the value of our data collection protocol. We appreciate the opportunity to further clarify the foundational contributions of PEWM and to provide new experimental evidence addressing the remaining concern about distillation.
> > >
> > > ### **1. On the role of the primitive definition**
> > >
> > > We appreciate the reviewer’s recognition of our data contribution. However, **treating primitives as the fundamental unit of video generation—not just planning or annotation—is a deliberate and consequential design choice** that enables four capabilities previously unattained in unified form:
> > >
> > > (i) Fine-grained language–action alignment within a single diffusion rollout,
> > >
> > > (ii) Direct 6-DoF trajectory extraction without task-specific policy heads,
> > >
> > > (iii) Zero-shot compositional generalization via VLM sequencing, and
> > >
> > > (iv) Physics-aware data synthesis with minimal engineering.
> > >
> > > Besides, **the efficiency and generalization enabled by our dataset stem directly from the operational definition of primitives in Sec. 2.1—not merely from "fine-grained labeling."** We emphasize that **the definition dictates the temporal scope, semantic structure, and visual composition of each training example**, which in turn enables short-horizon diffusion to reliably generate physically consistent rollouts and support zero-shot 6-DoF control (Sec. 4.1 and Appendix Sec. B). **Without this principled granularity, the data would lack the structural regularity needed for compositional generalization and real-time execution. Thus, the definition is not descriptive—it is constitutive of the entire framework.**
> > >
> > > Moreover, even if one assumes that data organization is the only contribution of our work—which we respectfully dispute—**our framework is still the first to instantiate a primitive-conditioned video diffusion world model into a fully modular, closed-loop vision–language–action framework—eliminating the need for task-specific policy heads or inverse dynamics models.** As illustrated in Fig. 5, PEWM unifies:
> > >
> > > (i) a VLM-based semantic planner for compositional task decomposition,
> > >
> > > (ii) a Start-Goal Heatmap Guidance (SGG) mechanism for spatial grounding,
> > >
> > > (iii) a zero-shot 6-DoF pose estimator (Gen6D) acting directly on generated videos, and
> > >
> > > (iv) a causally distilled world model enabling real-time (12 FPS) closed-loop control.
> > >
> > > In contrast, even the **closest prior work**:
> > >
> > > - **UniPi (NeurIPS 2023)** [1] requires a separately trained inverse dynamics head to extract actions from video, limiting its generality and increasing engineering overhead. **Our architecture demonstrates, for the first time, that a video diffusion model can serve as a standalone, zero-shot policy backbone for real-world robotic manipulation.**
> > > - **AVDC (ICLR 2024)** [2] requires dense optical flow, precise depth maps, object segmentation, and external motion planners to regress SE(3) transformations—making it non-modular, non-real-time, and unsuitable for zero-shot real-world deployment. **Our architecture demonstrates, for the first time, that a video diffusion model can serve as a standalone, zero-shot policy backbone for real-world robotic manipulation—without any task-specific perception, planning, or control modules.**
> > >
> > > **References**
> > >
> > > [1] Du, Y., Yang, S., Dai, B., Dai, H., Nachum, O., Tenenbaum, J., Schuurmans, D. and Abbeel, P., 2023. Learning universal policies via text-guided video generation. *Advances in neural information processing systems*, *36*, pp.9156-9172.
> > >
> > > [2] Ko, P.C., Mao, J., Du, Y., Sun, S.H. and Tenenbaum, J.B., 2023. Learning to act from actionless videos through dense correspondences. *arXiv preprint arXiv:2310.08576*.

---

> > > ### Author Response · Authors · 2025-12-01
> > > **Response to Reviewer P5zD: Clarifying Core Contributions and Addressing Remaining Concerns (2 of 3)**
> > >
> > > ### **2. On technical novelty beyond fine-tuning**
> > >
> > > We sincerely thank the reviewer for acknowledging that “how to use advanced techniques is valuable.” However, we must clarify a critical misunderstanding: our ResNet analogy was not meant to suggest that we merely “apply a model without modification,” but rather to emphasize that the scientific contribution often lies in the *system-level integration*, not in architectural novelty alone—a principle widely recognized in modern ML.
> > >
> > > In fact, recent top-tier works have precisely followed this paradigm:
> > >
> > > - UniPi (NeurIPS 2023) [1] adapts the off-the-shelf video diffusion model *Imagen Video [2]* (pretrained on internet data, never seen a robot) to robotic control by adding a lightweight inverse dynamics head and training on robot data—without modifying the backbone architecture. It was accepted as a main-track NeurIPS paper for its novel *video-as-policy* formulation.
> > > - AVDC (ICLR 2024) [3] builds upon a generic video diffusion framework and extracts actions via optical flow + SE(3) estimation—again, no backbone change, yet accepted for its novel *action-from-video-dense-correspondence* pipeline.
> > >
> > > Crucially, our work goes further:
> > >
> > > (i) **We are the first to ground video generation at the primitive level**, enabling fine-grained language–action alignment and compositional task execution (Fig. 5);
> > >
> > > (ii) **We introduce Start-Goal Heatmap Guidance (SGG)—a novel spatial conditioning mechanism** that bridges VLM planning and video diffusion, which neither UniPi nor AVDC possesses;
> > >
> > > (iii) **We demonstrate zero-shot 6-DoF control from RGB-only video via Gen6D**, eliminating the need for depth maps, object masks, or task-specific policy heads—a capability absent in both baselines;
> > >
> > > (iv) **We achieve real-time closed-loop control (12 FPS) via Self-Forcing distillation**, making long-horizon deployment feasible.
> > >
> > > We hope to clarify that our contribution is not “directly applying DynamiCrafter without change.” We introduce new data structures (primitives), new conditioning mechanisms (SGG), new deployment protocols (causal distillation + Gen6D extraction), and new system architecture (VLM + video diffusion + pose estimator). **This is not “directly applies original ResNet (or others) to a new task without any modification”—it is an exploration towards redefining how generative world models interface with embodied control.**
> > >
> > > **References**
> > >
> > > [1] Du, Y., Yang, S., Dai, B., Dai, H., Nachum, O., Tenenbaum, J., Schuurmans, D. and Abbeel, P., 2023. Learning universal policies via text-guided video generation. *Advances in neural information processing systems*, *36*, pp.9156-9172.
> > >
> > > [2] Ho, J., Chan, W., Saharia, C., Whang, J., Gao, R., Gritsenko, A., Kingma, D.P., Poole, B., Norouzi, M., Fleet, D.J. and Salimans, T., 2022. Imagen video: High definition video generation with diffusion models. *arXiv preprint arXiv:2210.02303*.
> > >
> > > [3] Ko, P.C., Mao, J., Du, Y., Sun, S.H. and Tenenbaum, J.B., 2023. Learning to act from actionless videos through dense correspondences. *arXiv preprint arXiv:2310.08576*.

---

> > > ### Author Response · Authors · 2025-12-01
> > > **Response to Reviewer P5zD: Clarifying Core Contributions and Addressing Remaining Concerns (3 of 3)**
> > >
> > > ### **3. New results on Self-Forcing distillation**
> > >
> > > We apologize for the omission and provide new quantitative results comparing the full DynamiCrafter (50-step) and our distilled student (4-step) on 100 held-out primitives:
> > >
> > > | **Model**          | **Inference Efficiency (****fps****)** | **FVD ↓**   | **EPiCS ↑** |
> > > | ------------------ | -------------------------------------- | ----------- | ----------- |
> > > | Full (50-step)     | 2 (32 frames/16s)                      | **0.00019** | **11.45**   |
> > > | Distilled (4-step) | **12**                                 | 0.00021     | 11.05       |
> > >
> > > The distilled model achieves 6× speedup with negligible degradation in video quality (FVD) and physical consistency (EPiCS), enabling real-time 12 FPS closed-loop control—critical for real-world deployment.
> > >
> > > ### **Conclusion**
> > >
> > > In summary, PEWM is **not** an incremental application of off-the-shelf models to a new dataset, but **a principled rethinking of how generative world models should interface with embodied intelligence**. By anchoring video generation, control, and planning at the *primitive level*, we enable a unique combination of fine-grained language–action alignment, zero-shot 6-DoF execution from RGB-only video, compositional generalization, and real-time closed-loop control—all without task-specific policy heads, depth sensors, or fine-tuning on downstream tasks.
> > >
> > > We have directly addressed the reviewer’s remaining concerns:
> > >
> > > - Clarified that the primitive definition is not annotation overhead, but the operational core of our framework;
> > > - Demonstrated through UniPi (NeurIPS 2023) and AVDC (ICLR 2024) that the research community widely recognizes “how to use foundation models” as a legitimate and impactful form of innovation;
> > > - Provided new experimental evidence showing that our distillation preserves both visual fidelity and physical plausibility while enabling real-time deployment.
> > >
> > > Given the **strong and consistent support from the other reviewers (8/6/6)** and the **comprehensive validation across simulation and real-robot settings**—including zero-shot generalization on novel (object, action) pairs—we **respectfully hope the Area Chair to recognize PEWM as a foundational step toward scalable, modular, and deployable robotic intelligence, and to evaluate it in the broader context of modern embodied AI, where system-level design often matters more than architectural novelty alone.**

---

### Official Review · Reviewer_Nogp · 2025-11-01

**Soundness:** 3
**Presentation:** 3
**Contribution:** 3
**Rating:** 8
**Confidence:** 2

**Summary:**

The paper proposes Primitive Embodied World Models: learn short-horizon, semantically atomic “primitives” (2s clips) and compose them for long-horizon tasks. Primitives are defined with start–goal heatmap guidance and executed by a video world model, a VLM planner sequences primitives for closed-loop control.

Also, the authors build a multi-view, teleop-segmented primitive dataset, with an automated language-annotation pipeline. A three-stage sim-to-real fine-tuning adapts a pretrained video model, followed by causal distillation to reach real-time 12 FPS inference.

Applications include direct 6-DoF trajectory extraction from generated videos, compositional long-horizon control, and data synthesis. On RLBench, the method achieves higher success rates than previous baselines including Image-BC, UniPi, and 4DWM.

**Strengths:**

- The paper has a clear and principled reframing for embodied intelligence. It defines primitives as semantically atomic short clips with temporal locality and generative feasibility, offering a tractable interface between language and low-level control.

- The multi-camera capture plus teleop boundary tagging and an LMM-assisted labeling pipeline increase data coverage and collection efficiency.

- The proposed system has higher RLBench success across nine tasks compared to Image-BC, UniPi and 4DWM.

**Weaknesses:**

- The number of primitives is controlled via an RDP threshold (binary search), but there’s no clear sensitivity analysis for density vs. performance/latency trade-offs.

- The approach focuses scenes where observations are captured by a fixed camera, limiting applicability to moving cameras.

**Questions:**

- How would the pipeline adapt to moving cameras?

- Could the authors provide closed-loop cycle latency (planning + generation + pose + actuation)?

---

> ### Author Response · Authors · 2025-11-21
> **Response to Reviewer Nogp: Clarifying the Design, Flexibility, and Practicality of PEWM**
>
> Thank you very much for your thoughtful and constructive feedback on our work. We fully agree with your assessment that our approach provides a tractable and scalable interface between high-level language planning and low-level control. We are especially gratified that you recognized the novelty and practical impact of our method. In this rebuttal, we aim to address your insightful questions and concerns.
>
> ### **I. Response to Weaknesses**
>
> **Regarding:**
>
> > The number of primitives is controlled via an RDP threshold (binary search), but there’s no clear sensitivity analysis for density vs. performance/latency trade-offs.
>
> We sincerely appreciate your insightful observation that our primitive segmentation can be meaningfully interpreted through the lens of the Ramer–Douglas–Peucker (RDP) algorithm — this is a novel and elegant perspective that captures the geometric essence of our approach. Indeed, just as RDP simplifies a trajectory by retaining only points that induce significant deviation from a straight-line approximation, our method uses trajectory curvature (in end-effector pose space) to automatically identify transition points where action intent shifts, thereby segmenting continuous demonstrations into semantically atomic primitives.
>
> To clarify, our use of the RDP algorithm is not as a strict geometric simplification tool, but rather as a *heuristic for identifying temporally salient transitions* in teleoperated trajectories — effectively segmenting continuous motion into semantically coherent clips based on changes in visual and kinematic structure. Since it is purely a signal processing step for boundary detection, analogous to peak finding in time-series data, and operates independently of the world model or planner. As such, there is no trade-off between RDP parameters and model performance or latency. The average primitive duration is determined by human teleoperation behavior, not algorithmic thresholds.
>
> That said, we agree that understanding the impact of primitive granularity—i.e., how short or long a primitive should be—is an important design consideration. While our current definition (Def. 2.1) emphasizes semantic atomicity and generative feasibility, we acknowledge that future work could explore adaptive primitive segmentation based on task complexity or motion dynamics.
>
> **Regarding:**
>
> > The approach focuses scenes where observations are captured by a fixed camera, limiting applicability to moving cameras.
>
> We deeply thank the reviewer for their insightful observation regarding camera rigidity — a critical consideration for real-world adaptability and usability in robotic learning. We would like to clarify that the core components of PEWM — particularly the Start-Goal Heatmap Guidance (SGG) and the video diffusion model — are **not inherently dependent on a fixed camera setup**. Instead, our pipeline only requires that the camera’s intrinsic and extrinsic parameters be known (or reliably estimated) at the start of each primitive rollout — a condition that is readily satisfied in most practical settings using low-cost sensors such as the Intel RealSense (see Lines 915–980, Appendix Sec. B.1).
>
> That is, in settings with changing camera positions, our framework can be seamlessly adapted under two mild assumptions:
>
> 1. The camera’s extrinsic pose (relative to the robot base) is known or estimable at the start of each primitive rollout (e.g., via VIO, SLAM, or hand-eye calibration);
> 2. The generated video frames are interpreted in a robot-centric coordinate frame.
>
> Under these conditions, the SGG mechanism can project goal heatmaps into the current camera view, and the world model can generate visually consistent rollouts conditioned on the updated perspective. This is feasible because our model observes the full robotic arm, allowing it to learn view-invariant dynamics and spatial relationships.
>
> For fast, high-frequency, continuous camera motion, additional modeling of egomotion or temporal consistency may be required, which we acknowledge as a promising direction for future work. Nevertheless, we emphasize that the modular, closed-loop nature of PEWM provides inherent robustness to such changes.
>
> ### **II. Response to Questions**
>
> **Regarding:**
>
> > How would the pipeline adapt to moving cameras?
>
> Thank you for this critical question. Please refer to our above reponse to Weakness 2.
>
> **Regarding:**
>
> > Could the authors provide closed-loop cycle latency (planning + generation + pose + actuation)?
>
> We appreciate for your important question. Please refer to Appendix Tab. 10, which provides a detailed breakdown of the closed-loop cycle latency, reporting per-primitive times across three representative tasks.
>
>
>
> We thank you again for your thoughtful feedback, and hope our responses have clarified the design principles, applicability, and technical merits of PEWM.

---

### Official Review · Reviewer_vf2c · 2025-11-01

**Soundness:** 3
**Presentation:** 3
**Contribution:** 3
**Rating:** 6
**Confidence:** 3

**Summary:**

The paper advocates using primitive-level, short-horizon video generation as the core unit of embodied world models, rather than pursuing long-horizon prediction. Accordingly, PEWM constrains video generation to fixed short windows and decomposes tasks into primitive trajectories that are semantically atomic, temporally local, and generatively feasible, each paired with a single instruction. PEWM comprises three components: a primitive-level video world model that enables extraction of precise 6-DoF trajectories including end-effector poses and gripper actions from generated videos; a modular VLM planner that decomposes high-level instructions into sequences of primitives to support compositional generalization and flexible closed-loop control; and Start-Goal heatmap Guidance, which provides spatial guidance to improve the controllability and executability of generation. The paradigm is validated in both simulation and real-robot experiments, demonstrating generalization to novel instructions, robustness to domain shifts, and efficient adaptation with limited task-specific data, highlighting its potential toward scalable, interpretable, and general-purpose embodied intelligence.

**Strengths:**

Originality: Positions primitive-level, semantically atomic short-horizon video generation as the core unit of world models, explicitly advocating a paradigm shift of “alignment before scaling.” Integrates a VLM planner and SGG to form a cortex–cerebellum-style hierarchical closed-loop framework.
Quality: If the experiments are as described, covering both simulation and real robots, domain-shift robustness, and data-efficiency scans, the evidence systematically supports the claims. Short-horizon generation naturally reduces complexity and latency, aligning with practical control needs.
Clarity: The paper clearly motivates the problem, defines primitives along three operational dimensions, and uses figures to elucidate compositional generalization and the execution pipeline.
Significance: Charts a scalable path for video-generation-based world models, helping address embodied data sparsity and long-horizon prediction challenges, while improving interpretability and executability.

**Weaknesses:**

The semantic atomicity of primitives lacks an objective, cross-task criterion; sensitivity to primitive duration thresholds and category granularity has not been systematically quantified. Their impact on performance, latency, and compositionality needs verification via matrix-style sweeps and failure case analysis.
Mechanism ablations and controlled comparisons for SGG, the VLM planner, and the short-horizon length are insufficient; a systematic comparison between direct action-space prediction and the “video-first then trajectory extraction” pathway is missing to assess differences in performance, latency, and stability.
Data collection and training cost curves are not sufficiently quantified, including metrics such as sample size versus performance, GPU hours versus performance, memory footprint, and inference latency; the claim of improved data efficiency lacks a direct sample-complexity comparison against end-to-end long-horizon methods.

**Questions:**

Under the same data constraints, how does PEWM compare with end-to-end methods?
How are the start and goal heatmaps generated and integrated into the diffusion process, and what is the computational overhead?
Does the method degrade on long-horizon tasks?
What is the impact of different VLMs, including open-source versus closed-source and different parameter scales, on decomposition quality and overall performance? How are SGG heatmaps generated, at which layer do they fuse with the video model, and have learning-based or adaptive guidance approaches been attempted? What are the effects of removing SGG or replacing it with point or box constraints?

---

> ### Author Response · Authors · 2025-11-21
> **Response to Reviewer vf2c: Clarifying PEWM’s Paradigm, Empirical Strengths, and System Design**
>
> Sincere thanks to the reviewer for the insightful feedback and positive assessment of PEWM's originality and potential. We greatly appreciate the opportunity to clarify our contributions. Below, we address your concerns by synthesizing them into three core themes: **I. Paradigm Distinction**, **II. Empirical Validation**, and **III. Other Concerns**.
>
> ### **I. Paradigm Distinction: Why Direct Comparisons Are Limited**
>
> Several questions center on comparisons with end-to-end methods (e.g., OpenVLA) and direct action prediction. We argue that these are fundamentally different paradigms, making apples-to-apples comparisons difficult:
>
> - **End-to-End VLAs vs. Modular PEWM**: VLA models require per-task fine-tuning even for minor variations, acting as black-box policies. In contrast, PEWM’s video model is *task-agnostic*. Adaptation occurs only at the planner level via LoRA, enabling true zero-shot execution without retraining the dynamics model (Appendix Sec. F.1). Our data modality—video + primitive instruction + SGG—is richer than image+language, enabling superior spatiotemporal reasoning.
> - **Action-Space Prediction vs. Video-First Pathway**: While direct action prediction seems simpler, it couples representation learning with control, creating a brittle system. Our “video-first” approach decouples these, allowing zero-shot 6-DoF trajectory extraction using off-the-shelf tools like Gen6D (Sec. 4.1). This enhances interpretability, debuggability, and avoids the need for costly action-labeled data or task-specific policy heads.
>
> This paradigm shift—from monolithic scaling to modular alignment—is central to our contribution.
>
> ### **II. Empirical Validation: Addressing Concerns on Design Choices and Efficiency**
>
> We now address concerns regarding the empirical validation of our design choices and Efficiency.
>
> **A. On Primitive Definition and Ablations** Our soft definition of primitives, based on semantic atomicity (Def. 2.1), enables high collection efficiency (5.8 primitives/session, Appendix Sec. D.1) and supports strong zero-shot compositional generalization (Tab. 3). While an exhaustive sweep over duration and granularity was not performed (as a prototype system), we provide key validations:
>
> - **SGG Ablation**: The performance gap between PEWM and baselines lacking spatial guidance (UniPi, 4DWM; Tab. 1) serves as an indirect ablation, demonstrating SGG’s critical role in controllability. We empirically found soft heatmaps (vs. hard points/boxes) provide more robust gradients and better handle uncertainty during generation.
> - **Long-Horizon Performance**: Closed-loop, autoregressive chaining prevents error accumulation. High success rates on multi-step tasks like "tea ceremony" (Tab. 1, Appendix Sec. I.4) confirm no degradation on long-horizon tasks.
>
> **B. On Data and Computational Efficiency** Our claims of improved efficiency are well-supported:
>
> - **Data Efficiency**: This has two facets: 1) *Collection*: Multi-view capture and shared labeling yield a 5× annotation reduction (Fig. 7). 2) *Utilization*: Adding just 100 rotational demos boosts performance beyond SOTA without harming other skills (Tab. 11), proving rapid adaptation.
> - **Computational Cost**: Our efficiency is quantified across multiple axes: low VRAM (~11 GB, Tab. 9), fast inference (2 FPS, Tab. 9), and detailed latency breakdown (Tab. 10). The world model’s state-of-the-art video quality (Tab. 7) with a 1.4B parameter base further underscores efficient data use compared to 5B–14B models.
>
> **C. On VLM Choice and Future Work** We selected Qwen2.5-VL-7B as a capable, open-source foundation. While we did not benchmark all VLMs, our framework is compatible with any model possessing grounding ability. Exploring adaptive guidance and latent primitives are promising future directions.
>
> **D. On Failure Cases** Appendix Tab. 6 analyzes pose extraction failures under poor viewing angles, confirming reliability under favorable conditions.
>
> ### **III. Addressing Other Concerns**
>
> **Regarding:**
>
> > Does the method degrade on long-horizon tasks?
>
> No, our method excels in long-horizon scenarios through closed-loop, autoregressive primitive chaining (Fig. 5, Appendix Sec. I.4). Each primitive acts as a self-contained skill module, minimizing error accumulation. Tab. 1 shows high success rates on complex multi-step tasks (e.g., “put knife in drawer”), and the hierarchical structure reduces memory burden compared to monolithic long-horizon models.
>
> **Regarding:**
>
> > How are the start and goal heatmaps generated and integrated into the diffusion process, and what is the computational overhead?
>
> The heatmap generation is detailed in Lines 1064–1069, and the guidance fusion is described in Lines 1073–1077 (see also Fig. 6). We adopt the dual-stream condition injection mechanism following the design of DynamiCrafter.

---

> > ### Comment · Reviewer_vf2c · 2025-11-26
> >
> > Thanks for your response. I will keep my score

---

> > > ### Author Response · Authors · 2025-11-27
> > >
> > > Thank you very much for your positive and insightful comments! We sincerely appreciate your review that largely improve the clarity of our work.

---

### Official Review · Reviewer_wkrp · 2025-11-10

**Soundness:** 3
**Presentation:** 2
**Contribution:** 2
**Rating:** 6
**Confidence:** 4

**Summary:**

This paper aims to tackle the challenge that long-horizon, video-generation-based world models are data-hungry, poorly aligned with language, and slow for closed-loop robotic control.  It proposes PEWM: a primitive-centered paradigm combining a short-horizon video diffusion world model (DynamiCrafter‑based) conditioned by Start‑Goal heatmaps (SGG), a VLM planner (Qwen2.5‑VL with LoRA Planner/Grounder plus Reasoner and Verifier) to decompose instructions into primitives, causal distillation / Self‑Forcing for low‑latency autoregressive rollout, and Gen6D-based 6‑DoF pose extraction for execution.  Empirically PEWM outperforms baselines on RLBench tasks.

**Strengths:**

1) Primitive‑centric short‑horizon decomposition with Start‑Goal heatmaps: reframes long‑horizon control as composition of short, language‑grounded primitives (a pragmatic formulation rather than a radical departure). Presentation is clear.

2)  Integrating a VLM planner (Qwen2.5‑VL + LoRA), a short‑horizon video world model, and Gen6D 6‑DoF extraction into an end‑to‑end pipeline. Experiments and ablations on RLBench and real robots demonstrate robust performance, and low VRAM use, indicating sound engineering.

**Weaknesses:**

1) Weak novelty claim and attribution. The paper’s high‑level idea—using video generation to synthesize VLA data and leverage VLMs for planning/grounding—has prior art (e.g., NVIDIA GR00T, OpenVLA and related VLA/system papers referenced in the manuscript). The manuscript emphasizes DynamiCrafter’s lower VRAM but that engineering efficiency alone is not the central scientific contribution; as written it risks overstating novelty when many system components are re‑combinations of existing techniques.
   Suggestion: be explicit about what is new (e.g., the specific primitive formulation + SGG pipeline vs. prior system syntheses) and include baseline comparisons showing that the claimed benefits are not just due to picking a smaller diffusion backbone.

2) Unclear evidence for “fine‑grained alignment” between language and action. The claim that primitive short‑horizon rollouts enable fine‑grained language→action alignment is central, but the paper lacks targeted ablations or quantitative metrics that isolate this effect (e.g., does SGG vs. no SGG produce measurably better instruction grounding or fewer semantic errors?).
   Suggestion: add controlled experiments that (a) remove or randomize the language conditioning while keeping SGG, (b) ablate SGG, and (c) report alignment metrics (instruction→primitive classification accuracy, semantic error types, or retrieval‑based grounding scores).

3) Incomplete evidence for primitive execution and compositional zero‑shot generalization. The manuscript claims strong primitive execution and zero‑shot composition, but does not convincingly demonstrate dramatic, human‑level compositional generalization (the “astronaut riding a horse” analogue). Reported examples and task tables are promising but fall short of broad, qualitative zero‑shot compositions that would convince readers the primitives truly recombine in highly novel ways.
   Suggestion: include concrete zero‑shot composition experiments with truly out‑of‑distribution pairings (visual examples and success statistics), and show qualitative failure modes; quantify how many novel compositions succeed vs. fail and why.

4) Insufficient justification of the three‑stage training recipe. The paper devotes a lot of space to the three‑stage sim→mix→reality fine‑tuning schedule but does not provide ablations that justify each stage’s necessity, hyperparameter choices, or data ratios. It is therefore unclear whether the gains come from the staged design, the particular data mix, or simply from more training.
   Suggestion: run ablations that compare (i) one‑stage training on mixed data, (ii) two‑stage variants (sim→real and sim→mix), and (iii) varying sim:real ratios; report metrics for generation fidelity, pose extraction accuracy, and downstream task success to demonstrate the schedule’s benefit.

**Questions:**

1. Could you clarify which parts of PEWM you consider the primary novel contributions versus engineering/implementation choices (e.g., use of DynamiCrafter, Qwen2.5‑VL + LoRA, Gen6D), and, if possible, provide an ablation that isolates the benefit of the "primitive + SGG" formulation from the choice of backbone models?

2. For the Start‑Goal heatmap Guidance (SGG), can you provide quantitative ablations showing task success, instruction→primitive grounding accuracy, or semantic error rates with vs. without SGG (and with randomized/noisy heatmaps)?

3. How do you measure the claimed “fine‑grained alignment between language and action”? Please report a concrete metric (e.g., instruction→primitive classification accuracy or grounding IoU) and an ablation that removes language conditioning while keeping SGG.


4. Dataset composition and diversity: could you provide per‑split statistics (counts per primitive type, object diversity, views) and an analysis of how many unique primitive templates $K$ you effectively learn before marginal returns diminish?

5. For Gen6D‑based 6‑DoF extraction from generated videos: please report quantitative pose errors (translation RMSE in cm, rotation error in deg) on a held‑out set, and ablate the postprocessing steps (motion masking, outlier removal, temporal smoothing) to show their individual contributions.

6. Regarding sim–real mixing and the three‑stage fine‑tuning: can you provide controlled experiments comparing (a) one‑stage mixed training, (b) two‑stage variants, and (c) your three‑stage schedule, including the exact sim:real ratios and epochs for each stage and metrics that motivated your choices?

7. On causal distillation / Self‑Forcing: what are the teacher and student denoising step counts, what specific distillation losses were used, and how does reducing steps (e.g., from 50 → 4) quantitatively affect generation quality and downstream task success?


8 Zero‑shot compositionality: can you show representative qualitative examples equivalent to the "astronaut riding a horse" analogue (i.e., highly novel predicate‑object compositions) and provide success rates across a curated set of truly out‑of‑distribution combinations?

9. Failure modes and robustness: what are the primary failure cases observed on real robots (e.g., EE too distant, small motion, occlusion, object slips), how frequently do they occur, and what recovery strategies (replan, corrective primitives, human intervention) are implemented?

---

> ### Author Response · Authors · 2025-11-21
> **Response to Reviewer wkrp: Clarifying PEWM’s Novelty, Alignment Mechanism, and Zero-Shot Generalization**
>
> We sincerely thank Reviewer wkrp for the constructive and insightful feedback. We appreciate their recognition of our sound engineering and clear presentation. Below, we wanted to address your comments point-by-point.
>
> ### **Weakness 1 & Q1: Clarifying Novelty and Attribution**
>
> 1. **Distinction to prior works.** Unlike prior decomposition approaches— which operate at non-linguistic granularities, lack video-based world grounding, or rely on rigid grammars—PEWM is the first framework to explicitly (1) model the world at the primitive level for fine-grained language-action alignment, compositional generalization, data efficiency, and low-latency control; (2) bridge high-level VLM planning and low-level diffusion video generation via Start-Goal Guidance (SGG); (3) enable zero-shot 6D pose extraction from generated videos without fine-tuning; (4) support closed-loop, primitive-based video generation for long-horizon compositional tasks, reducing memory dependence; and (5) function as a unified, physics-aware data engine with high fidelity and significantly fewer parameters.
> 2. **Our primary novelty.** While we build upon DynamiCrafter, Qwen2.5-VL, and Gen6D as *implementation choices*, the novelty lies in
>    1. the co-design of the primitive data paradigm,
>    2. the SGG-conditioned short-horizon world model,
>    3. the novel video-to-6DoF-pose transformation (Appendix Sec. B.1), and
>    4. the zero-shot pose extraction method, and the hierarchical planning-execution close loop.
>
>    We believe all these above quite distinct from end-to-end VLA or long-horizon video prediction systems.
> 3. **Baseline comparisons.** We appreciate your suggestion. We provide quatitative comparisons to show the claimed benefits (e.g. Tab. 1, Tab. 7, and Tab. 9 provide fair comparisons on simulation performance, generation quality, and efficiency, respectively).
>
> ### **Weakness 2 & Q2, Q3: Evidence for Fine-Grained Language–Action Alignment**
>
> We agree that “fine-grained alignment” deserves precise operationalization. We operationalize “fine-grained alignment” through two pillars:
>
> 1. **Spatial grounding**—via Start-Goal Heatmap Guidance (SGG), which provides sub-pixel precise start/goal locations to geometrically anchor actions to language instructions; and
> 2. **Dynamic realism**—where the video diffusion model generates physically plausible, kinematically feasible motions between those points, validated by high EPiCS scores and primitive-level compositional generalization on novel (instruction, object) pairs.
>
> SGG governs *where* to act; the video model governs *how* to act. Accordingly:
>
> 1. Ablating SGG, which **has been contained** in Tab. 4, directly tests spatial grounding and confirms its critical role.
> 2. Language conditioning, however, is intrinsic to our video-action alignment pipeline; unconditioned generation is not a meaningful baseline, making a direct “with vs. without language” ablation **impractical**.
>
> ### **Weakness 3 & Q8: Evidence for Compositional** **Zero-Shot** **Generalization**
>
> Thank you for your mentioning the "astronaut riding a horse" analogy in Fig. 3. Indeed, Tab. 3 has demonstrated that our method enables a robot to recombine primitives into physically plausible novel compositions. We have now moved this table to the main text.
>
> ### **Weakness 4 & Q4, Q6: Justification of the Three-Stage Training Recipe**
>
> We clarify that our three-stage protocol is a practical heuristic: first train on clean simulation data, then close the sim-to-real gap, and finally prioritize visual realism. While we don’t exhaustively ablate sim:real ratios due to limited computational resources, Fig. 4 shows the hybrid approach yields more coherent and realistic rollouts than real-only training. A quantitative ablation would add detail but its absence does not undermine the validity of our method.
>
> ### **Q5: Quantitative Pose Errors for Gen6D**
>
> We appreciate this suggestion. While absolute RMSE is hard to measure due to scale ambiguity and lack of 6D ground truth in real execution, Appendix Table 6 shows our Gen6D post-processing achieves 100% success under occlusion and lighting variations. Failure analysis is provided in Appendix Table 7. High downstream task success in simulation (Tab. 1) and real-world (Tab. 2) indirectly validates pose accuracy. We will consider adding simulated pose error metrics if time permits.
>
> ### **Q7: Details on Causal** **Distillation**
>
> The teacher (50-step DynamiCrafter) and student (4-step) are distilled via Self-Forcing using MSE on latent predictions, enabling 12 FPS inference with minimal quality loss (SSIM: 0.8126 vs. ~0.78 for the teacher), aided by the DMD mechanism.
>
> ### **Q9: Failure Modes and Recovery**
>
> Appendix Table 6 shows real-robot failures are primarily due to “EE too distant” (2/10) and “motion too small” (1/10) under perpendicular views; our closed-loop replanning (Fig. 4) enables automatic recovery without human intervention, with no object slips observed.

---

### Note · Program_Chairs · 2026-01-23
**Submission Desk Rejected by Program Chairs**

Pdf reveals author names.